# Concept transfer of synaptic diversity from biological to artificial neural networks

Martin Hofmann [1] ✉, Moritz Franz Peter Becker [2], Christian Tetzlaff[2,3] & Patrick Mäder [1,4,5]

Recent developments in artificial neural networks have drawn inspiration from biological neural networks, leveraging the concept of the artificial neuron to model the learning abilities of biological nerve cells. However, while neuroscience has provided new insights into the mechanisms of biological neural networks, only a limited number of these concepts have been directly applied to artificial neural networks, with no guarantee of improved performance. Here, we address the discrepancy between the inhomogeneous and dynamic structures of biological neural networks and the largely homogeneous and fixed topologies of artificial neural networks. Specifically, we demonstrate successful integration of concepts of synaptic diversity, including spontaneous spine remodeling, synaptic plasticity diversity, and multi-synaptic connectivity, into artificial neural networks. Our findings reveal increased learning speed, prediction accuracy, and resilience to gradient inversion attacks. Our publicly available drop-in replacement code enables easy incorporation of these proposed concepts into existing networks.

The field of machine learning and artificial intelligence has seen numerous advances in biologically motivated methods, leading to the development of key theories and concepts such as the McCulloch-Pitts cell[1], backpropagation learning[2,3], and convolutional neural networks inspired by the visual cortex[4–6]. Studies that build upon or revisit these foundational works frequently evaluate the biological plausibility of proposed methods, such as targetprop being more biologically feasible than backpropagation while producing similar performance[7–9] or exploring the absence of biologically inspired random backward connections in modern artificial networks[10]. Other studies propose methods based on novel models of artificial neurons[11], focus on new types of neural connections[12], propose binocular data processing[13], or study spiking neural networks with biologically inspired dynamics[14]. Hofmann and Mäder proposed synaptic scaling[15], an artificial neural network (ANN) training regularization method inspired by Tetzlaff et al.'s plasticity

rules of biological neural networks (BNNs)[16]. Blier et al. found that training with random learning rates over several orders of magnitude can improve robustness to hyperparameter variation[17]. However, direct compatibility with state-of-the-art artificial network architectures remains limited, and the scalability of biologically plausible learning methods is discussed as a fundamental challenge in machine learning[9].

From a neuroscience perspective, biological neural networks are highly complex structures consisting of a wide variety of neuron and synapse types. Decades of experimental neuroscience research have shown that synapses are diverse and dynamic. Their number, molecular composition, and morphology are constantly changing[18–20]. Such modulations lead to synapse- and neuron-specific synaptic plasticity, multi-synaptic connectivity between pairs of neurons, and spontaneous remodeling of synaptic connections. These changes result in specific adjustments to synaptic strength and structure, facilitate

[1]Data-intensive Systems and Visualization Group (dAI.SY), Technische Universität Ilmenau, Max-Planck-Ring 14, Ilmenau 98693 Thuringia, Germany. [2]Group of Computational Synaptic Physiology, Department for Neuro- and Sensory Physiology, University Medical Center Göttingen, Humboldtallee 23, Göttingen 37073 Lower Saxony, Germany. [3]Campus-Institut Data Science (CIDAS), University of Göttingen, Goldschmidtstraße 1, Göttingen 37077 Lower Saxony, Germany. [4]German Centre for Integrative Biodiversity Research (iDiv) Halle-Jena-Leipzig, Deutscher Platz 5e, Leipzig 04103 Saxony, Germany. [5]Faculty of Biological Sciences, Friedrich Schiller University, Fürstengraben 1, Jena 07745 Thuringia, Germany. ✉e-mail: martin.hofmann@tu-ilmenau.de

connections between multiple synapses and pairs of neurons, and lead to the spontaneous reorganization of synaptic connections. However, ANNs have not considered this synaptic diversity. We aim to investigate whether integrating the concept of synaptic diversity can improve the performance of ANNs and get an intuition on the underlying mechanisms.

In this paper, we study synaptic diversity when introduced into common ANN architectures. Specifically, we focus on three biologically inspired mechanisms (Fig. 1): diversity in synaptic plasticity, spontaneous spine remodeling, and multisynaptic connectivity. For each of these mechanisms, we propose a computationally lightweight implementation aiming at applicability to state-of-the-art ANN architectures. In an experimental setup with three of these architectures and three benchmark datasets, we evaluate each method separately and in combination and measure their effectiveness in terms of learning speed and model performance. Furthermore, we see that these mechanisms introduce a high degree of stochasticity into the networks. Given the assumption that such stochasticity could impede recovering training samples from the network, we measure the robustness of the proposed models to gradient inversion. Robustness

against gradient inversion, a form of adversarial attack, has garnered increased interest, particularly in scenarios involving decentralized learning from undisclosed data[21,22].

Learning is associated with synaptic plasticity, the ability of synapses to change their effectiveness in response to neuronal activity. Two well-established forms of synaptic plasticity are long-term potentiation (LTP) and long-term depression (LTD). However, the amount of potentiation and depression expressed at synapses depends on several factors, such as the type and frequency of neuronal firing patterns, but also on the brain region, neuron, and synapse type[23,24]. In addition, the location of synapses on the dendritic tree influences synaptic plasticity. Back-propagating action potentials are attenuated as they travel along the dendritic tree, making synapses distant from the soma less susceptible to potentiation than proximal synapses[25,26]. The ability of synapses to undergo activity-induced changes also depends on the history, size, and age of the synapse and neuron. Such history-dependent modulation of synaptic plasticity is termed metaplasticity[27]. Recent studies have shown that the stimulus-response and population coupling of individual neurons is variable[28–30]. While some neurons show stable responses to a given

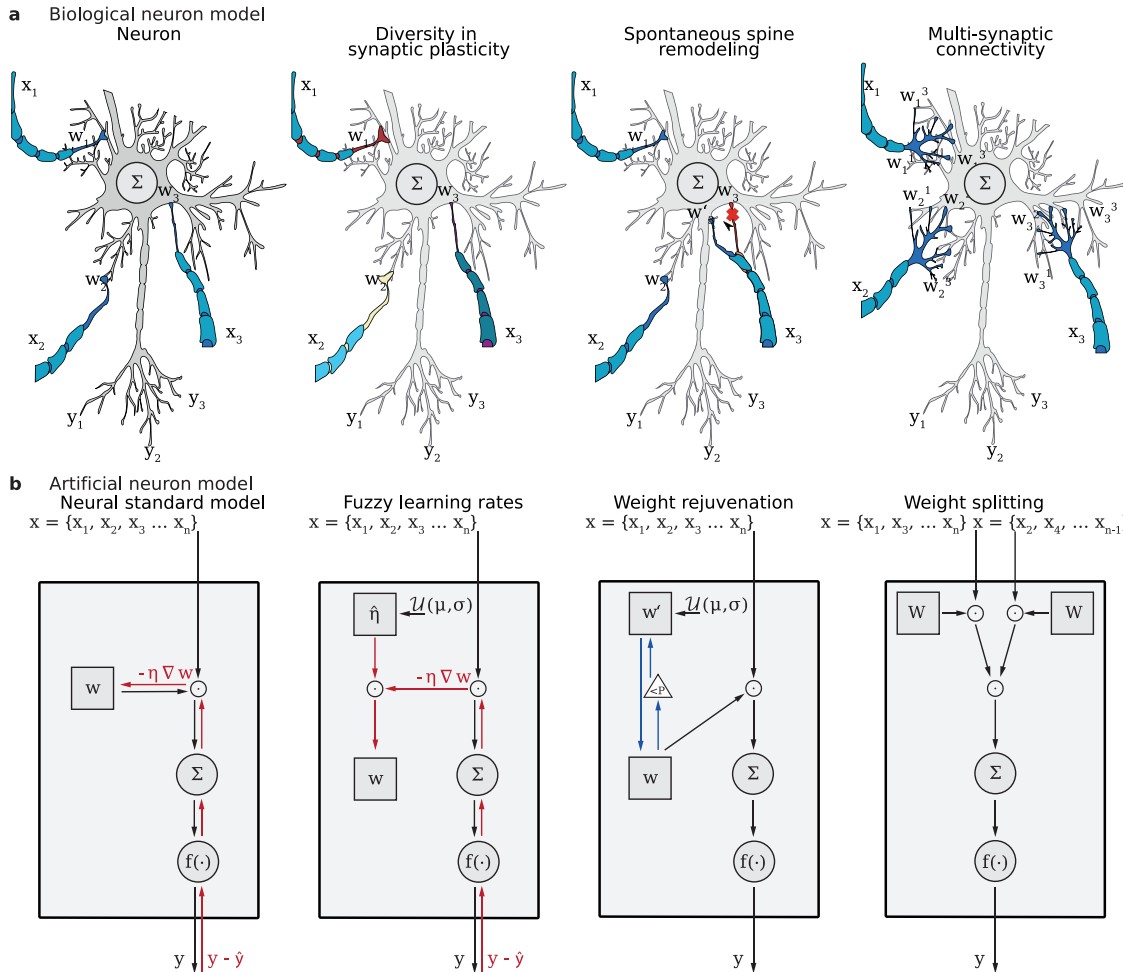

**Fig. 1 | Visualization of the examined synaptic diversity mechanisms and their respective artificial model. a, b** visualize the concepts that we have transferred from a simplified biological model to the artificial neuron. Accordingly, the figures show how these concepts correspond to functional elements in the artificial model. Here $x_i$ denotes the input of the presynaptic neuron, $w_i$ the synaptic weight, and $y_i$ the output of the postsynaptic neuron. $f(\cdot)$ denotes the activation function. Red arrows indicate the backpropagation path, and blue arrows indicate the process that overrides the weights. Circles denote operations, and squares denote

parameters. Black arrows indicate inputs and outputs. From left to right, the leftmost figure refers to a neuron, as most ANN models are interpreted today. The second illustration shows a neuron in which individual synapses have different learning speeds, highlighted by different color brightness. The third illustration shows small synapses subject to random reinitialization, representing spine remodeling and pruning. An apostrophe indicates the weight of the pruned synapse. The fourth illustration shows a neuron with multiple synapses between two neurons. The biological neuron is adapted from[139] (released into the public domain).

stimulus over days, others are highly dynamic[29,31,32]. It has been hypothesized that this variability results from different neuron-specific learning rates. This observation has functional implications; whereas neurons with high learning rates can flexibly learn new stimulus associations, less plastic neurons act as a stable, perturbation-resistant "backbone" of stimulus representations[29]. Thus, numerous mechanisms influence synaptic plasticity and, by this, the rate at which synapses change. In contrast, in ANNs the learning rate is typically fixed for all model parameters. Inspired by the biological concept, we propose different learning rates within a model by applying randomly generated constant factors to the gradients of network's synapses. This means that a randomly initialized constant for each of the trainable parameters of a network softens each learning step. We call this approach fuzzy learning rates (FL). This concept involves the introduction of a randomly initialized constant for each trainable parameter of a network, causing a disturbance in each learning step. Blier et al. observe that variation in the learning rates of artificial neurons benefits hyperparameter robustness[17]. We hypothesize that diversity in synaptic plasticity, realized as FL, can stabilize and regularize learning compared to the traditional approach. Other approaches like[33] research on the Incremental Delta-Bar-Delta (IDBD) algorithm illustrates the significance of adaptive learning rates in enhancing learning efficiency, a concept that resonates with the notion of FL in ANNs. The IDBD algorithm's capacity to adaptively modify learning rates according to input relevance bears resemblance to the suggested approach of introducing randomly generated constant factors to the gradients of network synapses, which serves to improve learning stability and regularization. Adaptive learning rates used by ref. 34 are similar to FL and are already observed to improve learning in linear systems. In contrast to the methodology presented by Hu et al.[35] that perform learning rate distribution modeling at a layer granularity, our approach implements this randomization at the individual synapse level. While they employ a beta distribution to regulate learning rates across entire layers utilizing Monte Carlo methods and dimensional reduction techniques, our FL approach directly introduces stochasticity at the synaptic level through random initialization of constants for each trainable parameter. This granular approach enables finer-grained control over synaptic plasticity but would render their distributional modeling approach computationally intractable due to the dramatically increased dimensionality of the parameter space. Specifically, where Hu et al.[35] compute redundancy among neurons using inter-neuron distances to modulate layer-wise learning rate distributions, our method draws inspiration from biological neural systems in which individual synapses exhibit variable plasticity[25,26]. The computational overhead of extending their distributional approach to individual parameters would scale quadratically with the number of parameters, making it prohibitively expensive for modern deep networks. Our synapse-level randomization achieves similar regularization benefits while maintaining linear computational complexity. This methodological distinction proves significant as it enables our approach to more accurately reflect the heterogeneous plasticity evident in biological neural systems[29], where individual synapses typically demonstrate varying degrees of plasticity based on multiple factors including spatial position and activation history. The granularity of our approach enables learning of stable "backbone" representations alongside more plastic components within the same layer-a property that would be difficult to achieve with layer-wise distributional approaches.

In biological neural networks, most excitatory synapses are formed by a dendritic spine to which an axonal terminal is attached. Dendritic spines grow, stabilize, and are pruned in an activity-dependent manner. Activity-dependent spine formation and pruning can depend on the correlation between neuronal activities (Hebbian-like), acting on a timescale of hours, or homeostatic-like, which controls network connectivity to reach a target activity level and acts on a timescale of days[19]. In addition to these activity-driven changes, dendritic spines are also subject to activity-independent, spontaneous remodeling and degradation[36,37]. Experimental studies have shown that the survival probability of dendritic spines is independently determined by their size and age[38,39]. Some spines are formed and pruned within days[38,40,41] while others are stable for months[39,40,42]. In addition, there is a positive correlation between spine size and synaptic strength[43]. Thus, as synapses experience potentiation of synaptic strength, their survival probability increases. Consistent with this notion, the proportion of persistent spines increases during development in mice[40]. Finally, while spine maturation is often associated with long-term potentiation, several studies have found that spines mature and form functional synapses even in the absence of synaptic activity[44–46]. Thus, dendritic spines are highly dynamic. In contrast, the connections between ANNs are typically stable entities that do not undergo spontaneous remodeling and pruning. Inspired by biological observations, we propose a model in which synapses spontaneously reinitialize depending on their current weight. We call this method weight rejuvenation. Other works, particularly in the field of Dynamic Sparse Training (DST), are not reinitializing new weights but eliminating them to increase generalization and also run-time properties and improve the hardware implementation ability to deliver key insights into[47,48] learning how neurons can be efficiently eliminated. The idea of DST[49] ranges from increasing the maximum model size for very large networks to reducing the number of floating point operations[50] calculating infrequent gradients per 1000 iterations. Also, reinforcement learning is helped by[51] that solves the problem with a control task, decreasing the computation costs while increasing the performance.[52] reinitialized synapses randomly with small weights and observed positive effects on learning. Weight rejuvenation comes with a continuous random noise similar to the noise used by ref. 52, where synapses are reinitialized if they do not contribute much to a result and are old enough to be reinitialized.

The ongoing formation and remodeling of spines and their respective synapses results in multi-synaptic connections between pairs of neurons, on average about 3–5 connections[19,53,54]. In contrast, the connections of ANNs are usually modeled by a single synapse. Multi-synaptic connections can have several functional implications. For example, it has been shown theoretically that the collective dynamics of multiple synapses can store information for a long time despite synaptic turnover[41]. We propose multi-synaptic connections that can be easily applied to existing ANN architectures. We call this method weight splitting. Connecting multiple synapses to a single input is expected to allow new activation patterns. For example, an input connected to a neuron via a positive and a negative weight will change the activation statistics. Multi-synaptic connectivity also distributes the gradient across multiple synapses in back-propagation, which can harden an ANN against gradient inversion attacks. Furthermore, in conjunction with diversity in synaptic plasticity, two neurons can be connected via multiple synapses, each governed by a different dynamic.

## Results

We systematically planned and executed a series of experiments to assess how biologically modified ANNs (biomod) perform on representative tasks in relation to benchmark ANNs. These experiments explore three proposed concepts to realize synaptic diversity in ANNs: (1) weight splitting (WS): Establishing multi-synaptic connectivity; (2) fuzzy learning rates (FL): Promoting diverse synaptic plasticity; (3) weight rejuvenation (WR): Enabling spontaneous remodeling of connections. The experiments are reported with different levels of optimization of networks' hyperparameters, such as network architecture, learning rate, and batch size. We also analyze memory and computational impacts of the proposed methods. Furthermore, we analyzed the network structure by observing changes in the Eigenvalue

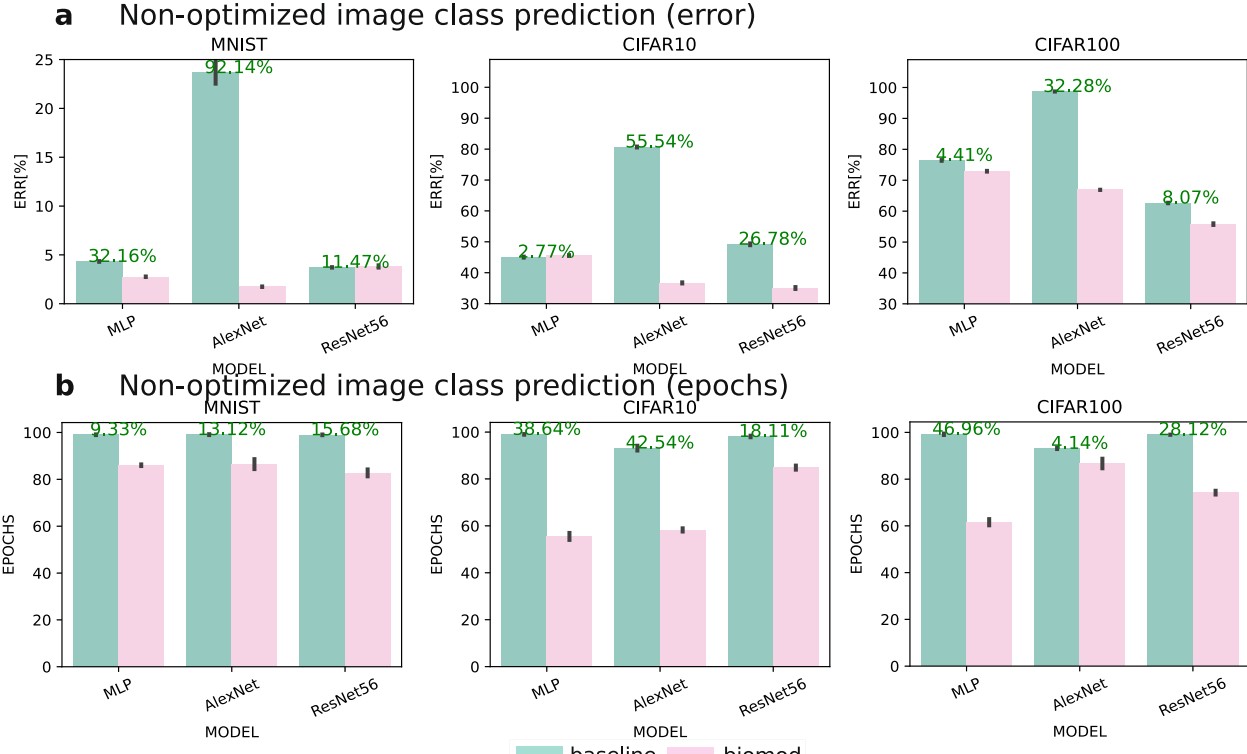

**Fig. 2 | Comprehensive performance evaluation of biologically inspired modifications (WS, WR, FL) on neural networks with default hyperparameters across MNIST, CIFAR10, and CIFAR100 datasets. a** Error rate comparison between baseline (red) models (MLP, AlexNet, ResNet56) and their biomod (green) counterparts, showing consistent error reduction across architectures and datasets, with particularly dramatic improvements for AlexNet on CIFAR10/100. **b** Training efficiency comparison showing fewer required epochs to reach peak accuracy in biomod, with improvements ranging from 9–47% reduction in training time. Black whiskers denote standard deviations.

spectrum of the optimization problem. Finally, we evaluate how biomod performs in a gradient inversion task to see how widely biomod could be useful. All results in the main text consider models with all three mechanisms. Please refer to the supplementary for analysis of the individual mechanisms.

**Efficiency of models with default hyperparameters**

This study is performed on MNIST[55], CIFAR10, and CIFAR100[56]. We assess the effect of non-optimized default hyperparameters on ANN while studying three architectures: a two-layer multilayer perceptron (MLP), an eight-layer AlexNet, and a 56-layer ResNet. The most relevant hyperparameters for all experiments, i.e., batch size, learning rate, gradient scaling rate $\tau$, the rejuvenation distance factor $d_{re}$, and the division factor $\Gamma$, have been set to default values derived via a search conducted with the MLP architecture trained on a 1% MNIST subset. We decided for merely 1% of MNIST to not scarify too much valuable training data and to prevent a very strong adaptation to a later analyzed problem. All remaining hyperparameters were set to the default value of the machine learning framework PyTorch[57].

Figure 2a shows a bar plot of the error rates on unoptimized hyperparameter settings. Our results show that the accuracy levels obtained with our proposed modifications are substantially higher than those of the baseline models (cp. Supplementary Table 1). We observe that the error rates of the AlexNet and ResNet architectures are unstable in the CIFAR10 and CIFAR100 configurations. Regarding the MLP experiments, our proposed modifications did not substantially improve the accuracy of the baseline model on the CIFAR10 dataset, with weight rejuvenation achieving the highest accuracy of 56.13%, compared to 55.00% for the baseline model. However, for the MLP–CIFAR100 configuration, weight splitting achieved the highest

accuracy of 28.33% (23.62% for the baseline model), and for MLP–MNIST, the combination of fuzzy learning rates, weight rejuvenation, and weight splitting achieved the highest accuracy of 97.25% (95.70% for the baseline model). For the AlexNet experiments, the baseline models produced unstable results with accuracies of 19.4% and 1.3% for the CIFAR10 and CIFAR100 settings respectively. However, we observed stable accuracies whenever weight splitting was applied, resulting in 63.42% and 33.71%, which were absolute 44.02% and 32.41% higher than the baseline models respectively. We also observed similar improvements in accuracy for the AlexNet–MNIST and ResNet56–CIFAR10 configurations, with absolute accuracy improvements of 21.45% and 14.42%, respectively. In other settings, we observed small absolute improvements ranging from 2% to 5%.

Figure 2b compares the learning speed between models with the proposed synaptic diversity and the traditional benchmark models (cp. also Supplementary Table 2). In all cases, the worst learning speed, i.e., the highest number of epochs, is observed for the baseline model. Another observation is that biomod did not overfit after achieving the highest accuracy, resulting in an overall superior area under the curve (AUC) (cp. also Supplementary Table 3). We also observed that ResNet56 exhibits clear overfitting behavior in its baseline configuration (FL = 0, WR = 0, WS = 0) and with only weight splitting enabled, showing accuracy decline after reaching peak performance on CIFAR10 and CIFAR100; however, this overfitting tendency is successfully mitigated when either fuzzy learning rates or weight rejuvenation is introduced, with the most stable post-peak performance achieved in configurations where both FL and WR are present (cp. Supplementary Fig. 1). From the AUC, we also conclude that all models reach a high accuracy fast but reach their highest accuracy late. We observe the lowest number of epochs and the highest AUC for a

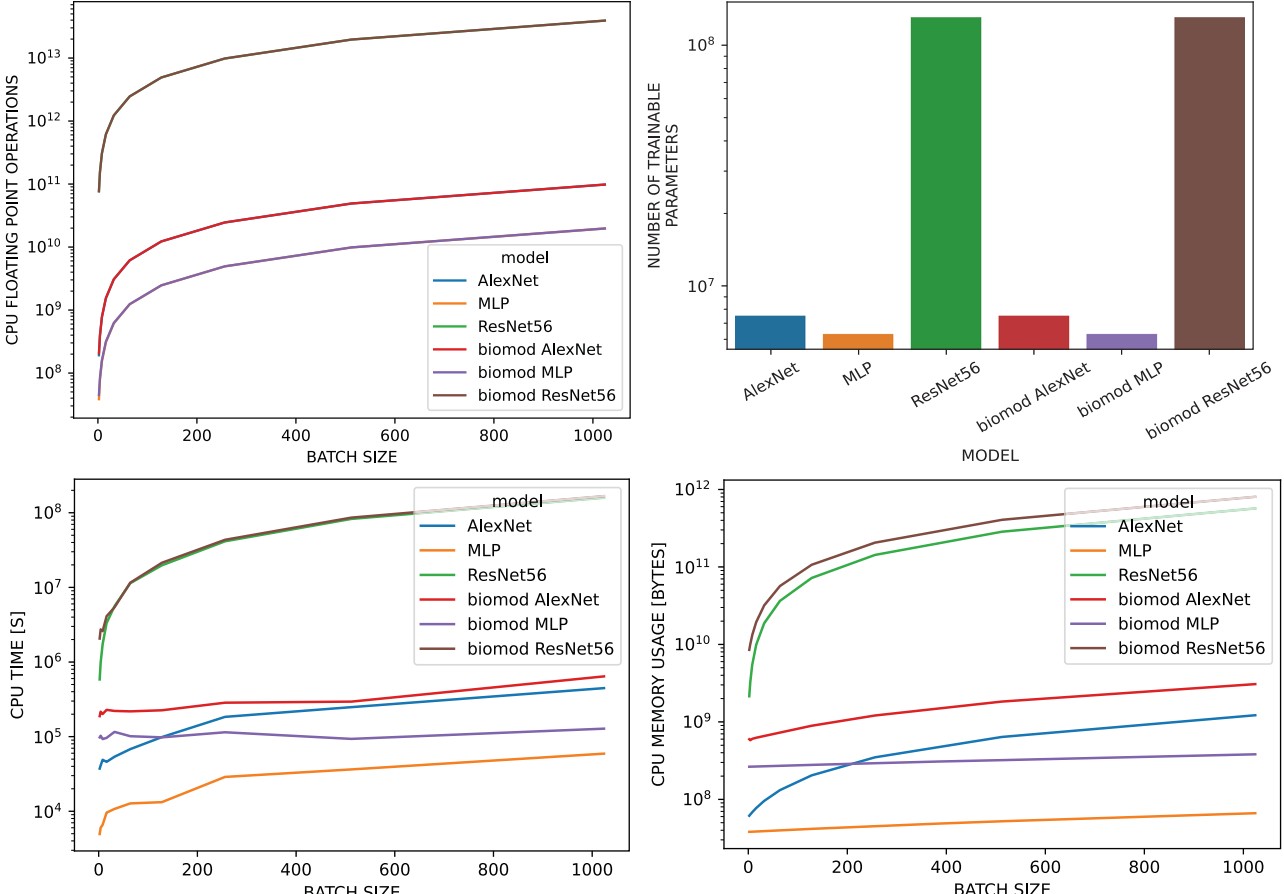

**Fig. 3 | Computational cost summary (cp. Supplementary Table 4) for AlexNet (blue, red), ResNet56 (green, brown), and MLP (orange, purple) with and without the biomod.** The evaluation includes profiling of memory usage, CPU time, and FLOPs during the forward pass, the backward pass, and the optimizer step. Computational costs and memory consumption are increasing from the MLP via AlexNet to the ResNet56 as to be expected. Models with biomod consist of twice as many parameters compared to the default versions, but their computational cost rises only marginally due to the methods' single application in the backward pass and diminishing impact with larger batch sizes. Top left: FLOPs vs batch size showing scaling behavior. Top right: Trainable parameter count comparison demonstrating the memory overhead of biological modifications. Bottom: CPU time (left) and memory usage (right) across batch sizes. While biomod contains twice the parameters, computational overhead remains modest, with impact diminishing at larger batch sizes.

combination of weight rejuvenation and weight splitting for the MLP–CIFAR10 and MLP–CIFAR100 configurations and with fuzzy learning rates for the MLP–MNIST setting. For AlexNet–MNIST, weight rejuvenation and weight splitting achieve the best learning speeds but fuzzy learning rates for AlexNet–CIFAR100. The best results are achieved across all trained ResNet56 models with a combination of all three methods. The same is true for the AlexNet–CIFAR10 configuration. However, we observe worse accuracies for the AlexNet–MNIST and CIFAR100 configurations, substantially deviating from the best results but still outperforming the baseline experiment when combining all three methods. For the MLP, the lowest number of training epochs in order to reach the best training performance across any dataset is observed when using Fl alone or WR combined with WS.

### Runtime and memory performance
In order to study the computational impact of the proposed methods, we evaluate CPU time, floating point operations (FLOPs), and memory consumption for AlexNet, ResNet56, and the MLP, each with and without the biomod methods applied (cp. Fig. 3). Our results show notable differences in computational efficiency across these models. The MLP exhibited the fastest performance, followed by AlexNet, while ResNet56 was the most demanding regarding time and memory. Although the biomod method doubles the number of parameters, its

computational impact is minimal as the additional parameters are only used once during the backward pass and are not trained. This effect is more pronounced in smaller networks like AlexNet and MLP, while it is almost negligible in ResNet. Consequently, the overall computational overhead remains relatively low compared to the actual computation, especially with larger batch sizes.

### Qualitative comparison of the loss landscape
Analyzing the loss landscape of neural networks is crucial due to several inherent challenges. The non-convex nature of the loss landscape, with its multiple local minima and saddle points, complicates the optimization process, making it difficult to find the global minimum[58]. Additionally, the high-dimensional parameter spaces of neural networks create vast and intricate loss landscapes that are hard to visualize and understand[59]. Moreover, saddle points, which are more prevalent than local minima in high-dimensional spaces, can significantly slow down training[60]. Understanding these aspects through loss landscape analysis can lead to the development of more effective optimization techniques and improved model performance.

The Hessian of a given optimization function provides a mathematical description of its curvature. It is a square matrix containing all second-order derivatives. This means that a Hessian Eigenmatrix of a neural network is a matrix with dimensions relative to the number of a

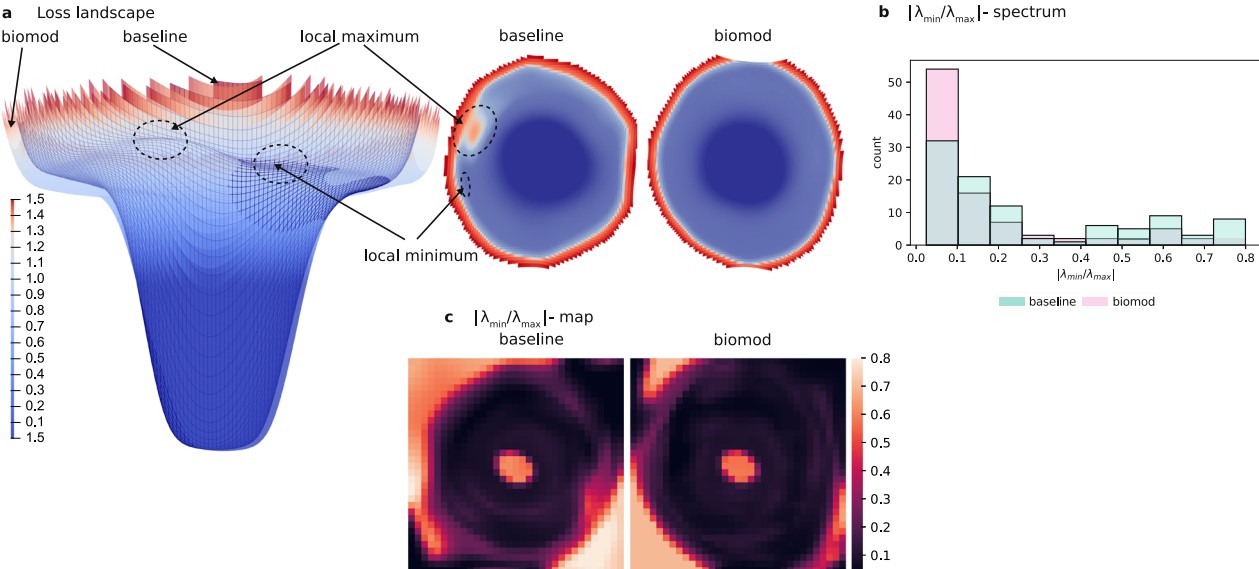

**Fig. 4 | Loss landscape visualization and Hessian Eigenvalue-MinMax ratio comparison between baseline and biomod ResNet18. a** The two plots represent the side view (left) and top view (right) of the loss landscape around the optimum, found by SGD training. Blue and red colors correspond to lower and higher loss values, respectively. The surface plot on the left shows loss values for the baseline model with mesh, while the surface without mesh shows biomod. **b** Comparison of the Hessian Eigenvalue-MinMax spectrum for the analyzed loss region between the baseline and biomod. Lower ratios indicate a more convex surface. **c** Hessian Eigenvalue-MinMax ratio for the plotted loss region in (**a**). Darker colors correspond to lower values, indicating a more convex surface, while lighter colors correspond to higher values.

model's parameters. Since it is not feasible to calculate these matrices for larger networks, LeCun et al.[61] used the power method, Taylor expansion, and the running average to calculate the quotient of the smallest and the largest Hessian Eigenvalue (condition number) without actually calculating the Hessian. Li et al.[62] employed that method to visualize the Eigenvalue MinMax spectrum of a network's parameters around the minimum. The quotient is especially interesting since the largest positive Eigenvalue determines the strengths of the convex curvature and the smallest (largest negative) Eigenvalue determines the strength of the non-convex curvature. If the absolute value of the largest eigenvalue is much greater than that of the smallest, the function can be considered primarily convex[63,64].

Figure 4a presents a loss landscape visualization that allows for a qualitative comparison of the ResNet18 architecture, which achieved an error rate of 3.76%, with FL+WR+WS refer to as biomod, which achieved an error rate of 3.64%, resulting in a relative reduction of 3%. Both loss landscapes are conical with a distinct shoulder step. The plotted area is limited to the region where the landscapes differ the most. The baseline model exhibits a less homogeneous area around the minimum, with local maxima and minima and an overall narrower area. Conversely, biomod shows a wider and flatter area. While these visualizations offer valuable insights, it is crucial to acknowledge their limitations in representing the vast complexity of high-dimensional spaces. To address this, we complement our analysis with the Eigenvalue-MinMax ratio spectrum across the loss landscape in Fig. 4b. For further observations see Supplementary Information: (Loss Landscape). We observe that both models have negative and positive Eigenvalues suggesting that the optimization problem is non-convex. However, the ratios are smaller for biomod with more than half of them belonging to the smallest bin. The median ratio is 0.17 for the baseline and 0.1 for biomod. Finally, in Fig. 4c, we present the Eigenvalue-MinMax ratio around the optimum for the presented loss landscape. The matrix shows a smooth surface with few local optima, indicating that the loss landscape is not highly non-convex. However, the Eigenvalue-MinMax-ratio reveals a higher ratio in the areas that appear flat in the loss landscape visualization, suggesting a non-convex

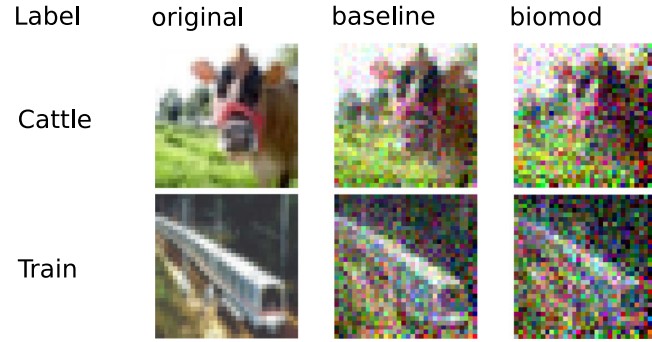

**Fig. 5 | Visual demonstration of enhanced privacy protection through gradient inversion resistance.** Comparison of original training images (left column) with reconstructed images from gradient information using baseline AlexNet (middle column) and biomod AlexNet (right column) after 100 epochs of training. Biomod significantly degrades reconstruction quality, evidenced by increased pixelation and loss of recognizable features, demonstrating enhanced protection against privacy attacks in distributed learning scenarios.

structure. Biomod is observed to have lower ratio values, indicating less non-convexity.

## Resilience to gradient inversion attacks

Federated learning is a growing field, e.g., due to the growing demand for medical data and the high demands that come with private data. These data are usually trained on distributed machines to preserve data privacy. However, a remaining vulnerability is the sharing of gradients over insecure connections, which has been shown to allow for the reconstruction of the private training data. To harden networks and training procedures against these reconstruction methods is a field of growing interest. Therefore, we also evaluated if the proposed models can also improve privacy in the described federated settings. Figure 5 visualizes the results of the gradient inversion experiment. Note that an error of 100% denotes a difference that is the same size as

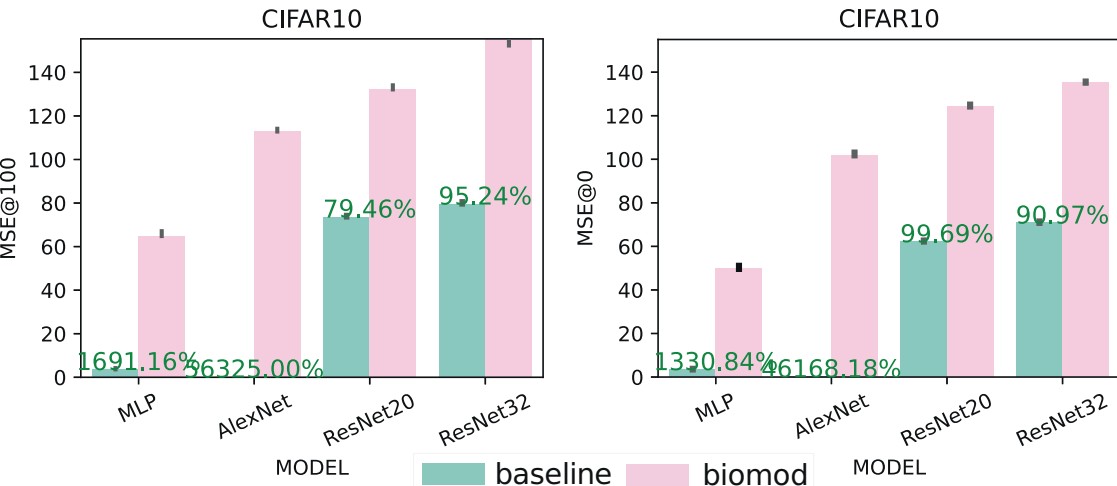

**Fig. 6 | Quantitative evaluation of gradient inversion resistance across neural architectures and biological modifications.** Bar plots compare mean squared reconstruction error (MSE) between baseline (green) and biomod (red) (MLP, AlexNet, ResNet20, ResNet32) on CIFAR10 dataset. Higher MSE values indicate better privacy protection, with biological modifications increasing reconstruction error by up to 56.3% in AlexNet and maintaining substantial improvements across all architectures. Black whiskers denote standard deviations.

the target value. WS and FL dramatically increase the reconstruction error and thereby privacy of training data in all setups. However, the combination of all methods yields the highest reconstruction errors. In some cases, WR is observed to have the least influence, failing to improve the reconstruction error. All results for the gradient inversion experiment are shown in Supplementary Table 5. We calculated the optimal reconstruction error based on the gradients of a batch. The reconstruction attacks are successful if they can achieve low reconstruction errors. The reconstruction errors are small, so we reported them in percentage. The attacks are very successful, especially for the untrained and unmodified MLP and the AlexNet. With errors of 3.47% and 0.22%, the reconstructed images show fine details as shown in Fig. 6, while maximum errors of 55.53% and 101.79% are reached for the untrained biomod. We observe the most significant reconstruction errors of 135.35% and 155.43% for bimod (all methods) untrained and trained ResNet32 architecture compared to 70.78% and 79.61% in the baseline case. We also observed that weight splitting alone improves the MSE in most cases, even in the untrained cases. FL alone does not improve the MSE as it is only applied during training.

## Task performances with optimized hyperparameter models

Here, we test several optimized models on four image classification benchmarks and two time series prediction benchmarks, spanning a wide range of ANN applications. We study four popular CNN architectures and a transformer for image classification and a mixture of two RNN, one CNN as well as a transformer architecture for time series forecasting. These models are, in most cases, already strongly adapted to the utilized benchmarks and yielding high performance. In order to gain representative results, we purposely chose those highly tuned models and evaluate how their performance changes when using the proposed methods.

Figure 7a shows the prediction performance of trained models with tuned hyperparameters using all proposed synaptic diversity methods in combination, along with the relative improvements over the respective baseline model (cp. also Supplementary Table 6). We observe reduced error rates across all datasets and models ranging from 0.02% to 19.64%, with a maximum observed standard deviation of 0.012%. WResNet28 yields the lowest error rate for CIFAR with a relative improvement of 0.021% for CIFAR10 and 0.24% for CIFAR100, respectively. Note that these results on CIFAR are substantially

improved compared to the original publication of the architecture (cp.[65] 3.02% CIFAR10 and 16.58% CIFAR100).

Figure 7b shows prediction performance for trained recurrent neural networks, i.e., LSTM and GRU, and a convolutional FDN[15] used to forecast time series (cp. also Supplementary Table 7). We observe positive effects on the normalized mean square error in all studied architecture-benchmark configurations, ranging from 3% to 26%. We further observe that the convolutional FDN architecture yields the lowest error among the baseline trainings and still is being improved when adding FL and WR (cp. Supplementary Table 7).

## Additional observations

We conducted various analyses on various aspects of our biologically inspired approaches, examining their impacts on neural network performance and internal dynamics.

- Our experiments on different fuzzy learning rate sampling strategies showed that uniform distribution resulted in the strongest improvement (Cohen's $d = 1.9$ for CNN, 0.84 for MLP) with optimal $\tau \approx 0.077$. All distributions significantly outperformed baseline. See Supplementary Figs. 2 and 3 and Supplementary Information: Fuzzy Learning Rates Sampling Distributions.
- Implementing Dale's principle by inserting excitatory, inhibitory, or mixed Neurons. Our observation resulted in 5–10% mixed neurons to be critical for learning stability. Here, we analyze how models change by integrating Dale's principle with FL, WR, and WS techniques. See Algorithm and weight analysis in Supplementary Information: Dale's Principle. Here, we present the main observations. Method combinations systematically transformed distributions, with biomod configuration achieving 98.17% accuracy. PCA revealed distinct clustering patterns in weight space. See Supplementary Figs. 4–6. Combined biomod produced highest peaks with lowest minima and broader, flatter basins, enabling smoother optimization paths. See Supplementary Figs. 7 and 8.
- Our analysis of catastrophic forgetting reveals significant performance differences across four methodologies: Continuous Backpropagation (CBP), L2 regularization, biologically motivated modulation (biomod, combining Fuzzy Learning, Weight Rejuvenation, and Weight Splitting), and biomod with CBP. The biomod and biomod+CBP approaches demonstrated

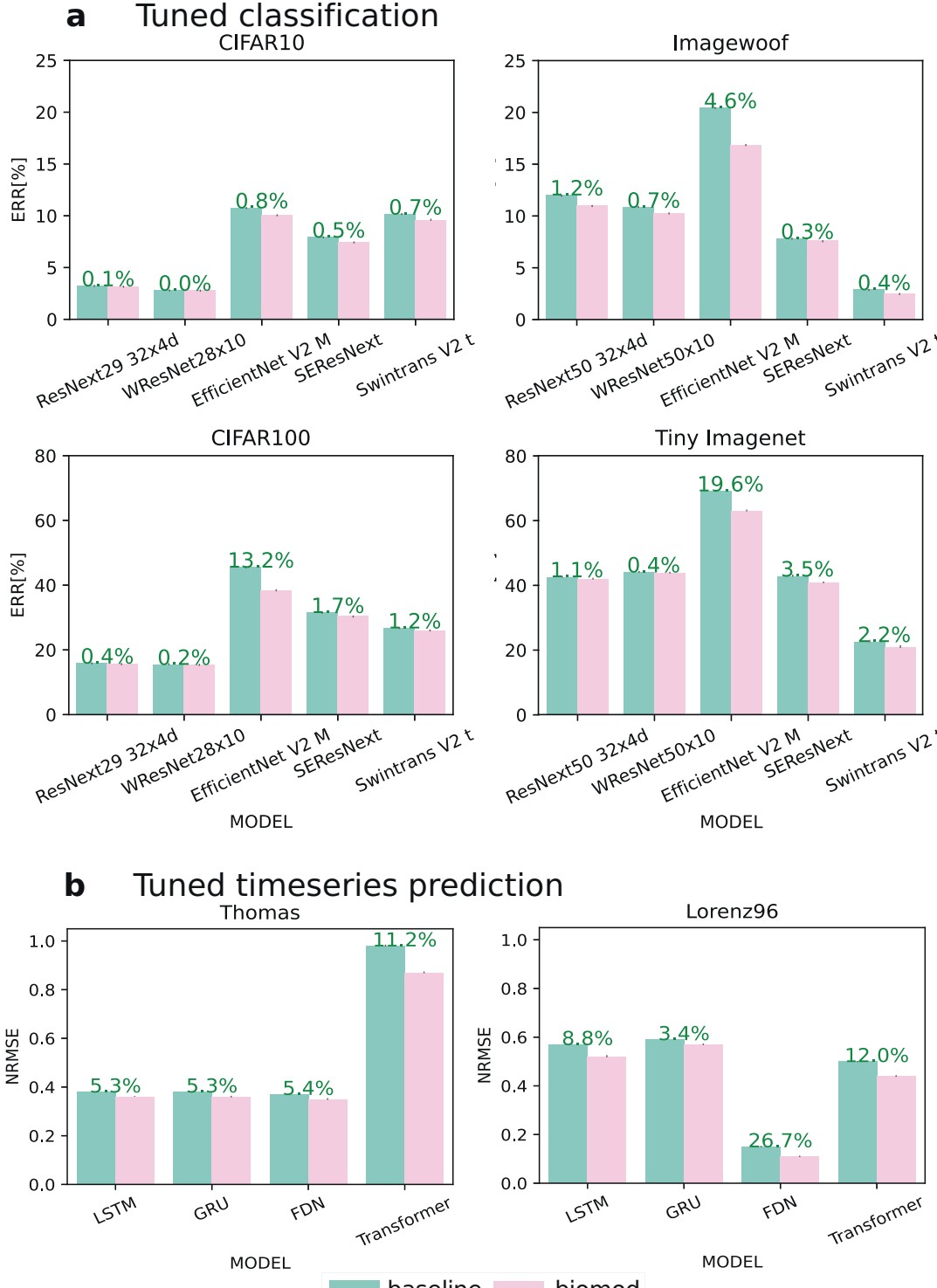

**Fig. 7 | Comprehensive performance evaluation of the baseline(green) and biomod (red) with optimized hyperparameters across diverse tasks.** **a** Classification error rates on four image datasets (CIFAR10, ImageWoof, CIFAR100, Tiny ImageNet) comparing state-of-the-art architectures (ResNext, WResNet, EfficientNet, SEResNeXt, Swintrans V2) with their biomod counterparts, showing consistent error reductions of 0.1–19.6%. **b** Time series prediction performance on Thomas and Lorenz96 benchmarks using various architectures (LSTM, GRU, FDN, Transformer), demonstrating NRMSE improvements of 3.4–11.2% through biological modifications. Small black whiskers denote standard deviations.

accelerated early learning and superior peak performance, exceeding 90% accuracy while maintaining stability throughout training. Notably, biomod+CBP achieved a maximum of 90.8% accuracy around iteration 2350, suggesting enhanced learning capacity with reduced variability compared to other methods

(cp. Supplementary Fig. 9). Detailed experimental procedures, including network architectures and hyperparameters based on implementations from[52] and comprehensive performance analysis, are presented in Supplementary Information: Catastrophic Forgetting.

## Discussion

Our methods unify structural and functional plasticity in neural networks, providing a further glance at variability and continuous rewiring in brain-like networks. This approach aligns with and extends previous work on stochastic plasticity[66–69]. The core of our methods rests on three key concepts: fuzzy learning rates, weight rejuvenation, and weight splitting.

Our observation that fuzzy learning rates (FL) positively influence learning speed, stability, and yield more optimal parameters aligns with observations in biologically observed networks where noise positively activates learning[70,71]. Although these observations have been made primarily in spiking neural networks, our findings do not contradict the assumption that random noise can benefit learning. The variability introduced by FL mirrors not only biological processes but also confers computational advantages, as suggested by ref. 72, potentially improving performance and stability through nonuniform initialization.

The weight rejuvenation mechanism in our model supports the genetic findings of Schuster et al.[73], who showed that synaptic growth is accompanied by changes in synaptic protein levels. This mechanism enables continuous adaptation and growth of synaptic connections, mirroring the biological resilience observed in self-assembling networks proposed by Plantec et al.[74]. It allows for rapid adaptation to new information while maintaining overall network stability, aligning with findings by Grooten et al.[75] on adapting to noisy and shifting environments.

Weight splitting introduces synaptic redundancy, allowing for both long-term stable representations and rapid adaptation. This aligns with the SpikePropamine framework[76] and the STDP-driven rewiring model[77], enabling both stable long-term memories and quick incorporation of new information. By duplicating neurons and summing them before activation, we introduce redundancy that may highlight critical features, similar to ensemble methods[78–84]. Regarding Performance and Adaptability, we tested our methods on various architectures under both optimized and default hyperparameter settings. All architectures benefited from our methods, with those not initially designed for CIFAR input sizes showing the most significant improvements. This suggests our approach introduces beneficial regularization, especially for less finely tuned architectures, and can mitigate overfitting tendencies in deeper architectures like ResNet56, as we observed (see Fig. 1). With default hyperparameters, we observed even more pronounced benefits, particularly when combining all proposed methods. For instance, ResNet56 showed an 8% improvement on CIFAR100 benchmarks. The observation of stronger effects in configurations with non-optimized, default hyperparameters is particularly relevant, suggesting increased resilience to poorly chosen parameters. This mirrors research on how the brain functions reliably in the presence of noise[85] and could apply to extremely noisy environment tasks in reinforcement learning[75].

Analyzing the Loss Landscape and Gradient Inversion Resistance, we found that our methods produce broader minima in loss functions with fewer local minima and lower Hessian Eigenvalue MinMax ratios, suggesting a smoother, less non-convex target function[60,64]. This explains the observed improvements in prediction accuracy and learning speed. Additionally, our methods provide resistance to gradient inversion attacks, a beneficial side effect of gradient weakening and splitting over different synapses. Additionally, our methods provide resistance to gradient inversion attacks, a beneficial side effect of gradient weakening and splitting over different synapses. Additional experiments (cp. Supplementary Loss Landscape) on smaller architectures for every combination of FL, WR, and WS, both with and without Dale's principle[86] (DP), confirm the main observations regarding broader minima and smoother descent. Dale's principle, which historically posited that neurons release the same neurotransmitter at all synapses[86], has been significantly challenged by

modern evidence of neuronal co-transmission[87–91]. As shown in the supplementary information, multi-transmitter neurons are now recognized as fundamental components of learning and memory circuits, with studies demonstrating that over 50% of terminals in pathways like the supramammillary-hippocampal exhibit dual transmitter capability[92] (see Supplementary Information: Dale's Principle). Notably, the imposition of Dale's principle consistently raised the final loss while narrowing the basin, yet it still yielded convergent solutions-an encouraging outcome given the usual challenges of training strictly excitatory inhibitory networks. Without Dale's principle, the triple combination of methods produced steeper and more direct trajectories through parameter space, underscoring the robustness of these approaches across a broader configuration set. These findings suggest that, although Dale's principle led to inferior performance here, its successful integration may ultimately expand biologically grounded modeling avenues in computational neuroscience.

Our analysis of weight distributions (see Supplementary Information: Weight Distributions) across different methodological settings reveals systematic patterns that support our loss landscape analysis. Synaptic weight distributions show distinct characteristics depending on the applied methods, with statistically significant deviations from the expected log-normal distribution. In this theoretical distribution, most weights are small, while a few are large, as previously observed by[15]. For the baseline configuration, we find strong deviations from log-normality, with a Shapiro-Wilk test yielding $p = 3 \times 10^{-6}$ and a $K^2$ test result of $6 \times 10^{-5}$. The distribution exhibits notable negative skewness (−1.29) and moderate positive kurtosis (1.14).

Individual methods induce systematic transformations in these distributions; notably, weight splitting substantially reduces skewness magnitude (-0.74) while normalizing kurtosis (0.23). Perhaps most intriguingly, biomod demonstrated superior performance in our loss landscape analysis-achieves peak accuracy (98.17%) while maintaining moderate deviations from perfect log-normality (skew = -0.44, kurtosis = -1.52). This finding reinforces the notion that optimal network performance does not necessarily demand strict adherence to log-normal distributions, aligning with our observations of broader minima in loss landscapes. Principal component analysis of weight spaces reveals distinct clustering patterns, particularly pronounced in configurations implementing multiple methods. Indeed, these configurations exhibit fundamental alterations in synaptic organization that correspond to the observed smoothing of loss landscapes. These distributional characteristics provide compelling evidence for the synergistic effects of our proposed methods, complementing our earlier findings on loss landscape geometry and gradient descent trajectories.

Catastrophic forgetting is the phenomenon where neural networks rapidly lose previously learned information when trained on new tasks or data distributions. Additional empirical observations on catastrophic forgetting (see Supplementary Information: Catastrophic Forgetting) indicate significant performance differences across methodologies. The biomod and biomod + DP (DP refers biomod trained incorporating Dale's Principle) configurations demonstrated enhanced early-phase learning acceleration, achieving accuracy levels of 87–88% within the initial 200 training iterations, surpassing the 86–87% baseline established by traditional approaches. This acceleration manifested with statistical consistency across multiple experimental runs ($p < 0.05$), indicating robust enhancement of initial learning dynamics. The intermediate phase revealed an informative hierarchical performance pattern: biomod maintained accuracy of 88–90%, while L2-regularized configurations stabilized at 87–88.5%. Notably, the performance gap remained statistically significant ($p < 0.01$) throughout this phase. During peak performance, distinctive spikes emerged: biomod+DP achieved maximum accuracy of 90.8% at iteration 2.350, exhibiting substantially reduced variance (±0.9%) compared to classical methods (±1.2%). The convergence analysis

revealed consistent performance increases of our biologically inspired approaches, with biomod variants stabilizing at 90–91% accuracy compared to L2's 89%. Our methods effectively unify structural and functional plasticity in neural networks, providing deeper insights into variability and continuous rewiring in brain-like networks. This approach aligns with and extends prior work on stochastic plasticity[66–69]. The core of our methodology rests on three key concepts: fuzzy learning rates, weight rejuvenation, and weight splitting.

## Limitations and future directions
Our study has limitations, including a restricted set of tested architectures and tasks. However, we aimed to cover relevant concepts such as convolutions, recurrence, batch normalization, and residual connections. Loss landscape visualizations provide only rough estimates of high-dimensional spaces, and conclusions should be considered cautiously. Future research should focus on developing stronger mathematical explanations for our observations, guided by work such as Kappel et al.[69]. We could create more realistic models of spine size dynamics[68], explore different types of priors[93] and astrocyte-mediated plasticity[94]. Our observation that implementing Dale's principle[95] leads to inferior results could be further explored by implementing heavily constrained plasticity[93]. Optimizing nonuniform sparse initialization[72] and dynamic weight adaptation[75] could further improve performance in challenging and changing environments. Our open-source Python package assists researchers in implementing these concepts, providing a flexible platform for them to extend. This tool can help bridge the gap between computational models and biological observations in neural plasticity research. Future work should study how our methods affect different parts of neural networks and how they interact with various optimization techniques, requiring expertise from both computational neuroscience and machine learning.

In conclusion, our synaptic sampling framework offers a promising approach to creating more robust, adaptable, and biologically plausible ANNs. Bridging computational principles with biological observations provides new insights into the mechanisms of learning, memory formation, and network robustness.

# Methods
We aim to bring and evaluate BNNs' core synaptic plasticity concepts, i.e., diversity in synaptic plasticity, spontaneous spine remodeling, and multi-synaptic connectivity, to ANNs. Therefore, we propose a formalization for each that aims to preserve the concept while being lightweight enough to be used as a plug-in replacement within common ANN architectures. We refer to these three formalizations as: fuzzy learning rates (FL), weight rejuvenation (WR), and weight splitting (WS).

## Fuzzy learning rates (FL)
Fuzzy learning rates aim to introduce diversity in synaptic plasticity to ANNs.

**Formalization.** We propose different synaptic learning rates $\hat{\eta}_{n,i}$ per synapse belonging to a neuron and affecting their corresponding weight $w_{n,i}$. Synapses are enumerated with $n = 0, 1, 2, \ldots$ and $i = 0, 1, 2, \ldots$, where $n$ denotes the post-synaptic and $i$ the presynaptic neuron. We denote the unbiased neural transfer function as $\phi_n = g(\sum_{i \in I} w_{n,i} x_i)$, where $x_i$ refers to the input of neuron $i$ and g refers to an arbitrary nonlinearity, e.g., a biologically motivated[96,97] rectified linear unit function (ReLU)[98–100]. For this reason, we also perform all experiments with biases initialized to zero. Accordingly, the learning rate of each synapse is realized as a constant random factor applied to its gradient. Thus, a typical gradient descent step changes from $w_{n,i,t+1} = w_{n,i,t} - \eta \nabla \phi_{n,i}$ into

$$w_{n,i,t+1} = w_{n,i,t} - \eta \nabla \phi_{n,i} \odot \hat{\eta}_{n,i}. \tag{1}$$

A factor $\hat{\eta}_{n,i}$ is randomly drawn from a uniform distribution per weight upon initialization of the network

$$\hat{\eta}_{n,i} \leftarrow \mathcal{U}(1 - \frac{\tau}{2}, 1 + \frac{\tau}{2}), \tag{2}$$

where $\mathcal{U}$ is the uniform distribution and $\tau$ is the gradient scaling rate. The runtime of this method is independent of the size of the input sample and is run once for all weights. However, the number of operations required to propagate the network increases linearly with the size of the model. Since biological neural networks exhibit diverse neural plasticity distributions (refs. [101–104]), we evaluated several distribution types (uniform, normal, log-normal, geometric, and beta) in a preliminary experiment (see Supplementary Information: Fuzzy Learning Rates sampling distributions). Finding no statistically significant differences in performance between distributions, we proceeded with the uniform distribution for our main experiments.

**Intuition.** Zhou et al.[105] found that optimizers with smoother gradient noise need more iterations to leave local minima because ADAM tends to favor sharper minima. They compared the ADAM optimizer to the SGD and observed that the SGD exits local minima faster due to heavier gradient noise, resulting in better convergence to lower minima. Neelakantan et al.[106] experimentally observed that larger networks generalize better when gradient noise is induced, and the local minima are wider. Methods that evaluate the sharpness of local minima and promote flatter minima are refs. [65,107].

## Weight rejuvenation (WR)
Weight rejuvenation aims to introduce random reinitialization of synaptic connections inspired by the spontaneous spine remodeling of BNNs.

**Formalization.** Weight rejuvenation means that a weight $w_{n,i}$ is reset to a random value with a certain probability, mimicking spinal purging and formation. More specifically, the smaller a weight becomes during a training process, the higher its probability of reinitialization. The Gaussian probability is derived via the commutative density function of the normal distribution:

$$_{\mathrm{re}} = 1 - \frac{1}{\sigma_{\mathrm{re}} \sqrt{2\pi}} \int_{-\infty}^{w_{n,i}} e^{-\frac{1}{2}\frac{t - \mu^2}{\sigma_{\mathrm{re}}}} dt, \tag{3}$$

where $\sigma_{\mathrm{re}}$ is the rejuvenation variance calculated with respect to the maximum value of a layer's synaptic weights and $\mu$ equals 0. Thereby,

$$\sigma_{\mathrm{re}} = |w_{\mathrm{max}}/d_{\mathrm{re}}|, \tag{4}$$

where $d_{\mathrm{re}}$ is the rejuvenation distance factor. For example, a rejuvenation distance factor of 1 means that the maximum synaptic weight of a layer $w_{\mathrm{max}}$ is reinitialized with a probability of ~16%. We later use a $d_{re}$ of 14 that shrinks the probability of rejuvenating the largest weight to zero. For example, weights <0.2 then have a probability of 39%, if the $w_{\mathrm{max}}$ is 1. After an initial phase, the number stagnates at a certain level, introducing further noise into the synaptic weights. The time consumption is independent of the size of the input data and is calculated once per training step for all weights. However, weight rejuvenation increases the total number of network operations linearly with the model size. Furthermore, trained ANNs are often characterized by relatively few large weights and most small weights[15]. Therefore, iterative rejuvenation of small weights is not expected to affect the current training progress but may help to explore new training directions.

**Intuition.** DropConnect randomly masks synaptic weights, resulting in noisy activation and improved generalization[108]. This method is related to weight rejuvenation, as it shows that noisy weights and even randomly masked weights can improve learning.

## Weight splitting (WS)

Weight splitting aims at incorporating multi-synaptic connectivity into ANNs, inspired by the observation that biological neurons often have multiple connections among each other (cp. Fig. 1).

**Formalization.** We implement WS by incorporating multiple inputs into each of the $N$ neurons. Here $\Gamma$ denotes the set of indices of the replicated neurons and its cardinality is interpreted as the number of connections between a pair of neurons. The division factor $\|\Gamma\|$ determines the number of inputs. The inputs are each multiplied by a weight and then aggregated using a transfer function followed by an aggregation function. We found that a plausible combination is an identity function as a transfer function and the sum as an aggregation function[41]. These resulting multiple linear units allow for varying weights per synapse and learning information faster than forgetting when WR is used. Similarly, the transfer function of a layer is denoted by

$$\phi_n = g\left(\sum_{\gamma \in \Gamma} \sum_{i \in I} f\left(w_{n+\gamma\lfloor \frac{N}{\|\Gamma\|}\rfloor, i}, x_i\right)\right) \forall\, 0 \leq n < \frac{N}{\|\Gamma\|}, \tag{5}$$

where $\lfloor \frac{N}{\|\Gamma\|}\rfloor$ is the distance of the accumulated synapse indices. The functions $f(\cdot)$, $g(\cdot)$ are activation functions. The parameters in the activation function $f(\cdot)$ are the input $x_i$ and its corresponding weight. Because $w$ is a continuous list of numbers, we need to group several weights for each input to produce one output. If $\Gamma$ is 1, meaning we are not using weight splitting (WS), each weight $w_n$ is multiplied by its corresponding input index $i$ for all $N$ output neurons. When we use weight splitting with a factor of 2, we reduce the number of outputs to $N$ divided by 2. The total number of weights remains the same, but now we skip every other weight when adding up the inputs for each neuron. This skipping is controlled by $\lfloor \frac{N}{\|\Gamma\|}\rfloor$, which helps us decide upon the weights to include in each neuron's calculations. To allow our method to work as a drop-in replacement, we need to increase the number of neurons after this operation. We, therefore, duplicate the neurons $\Gamma$ times so that the resulting copies are not the inputs of the aggregation in the next layer. This leads to changed connectivity in the network, possibly changing its behavior. WS increases the total number of operations of a network linearly with the model size, but independently of the input sample size.

We did not include Dale's principle because we wanted to reduce the parameter complexity of our evaluations, and we did not find any configuration where the training was not heavily impaled.

**Intuition.** Gated Linear Units[109] is a related concept that combines the identity function and a sigmoid function with a product aggregation function. The method provides a strong gradient over deep networks and performs feature selection in NLP tasks. However, more research is needed to find biologically plausible combinations.

## Experimental setup

We conducted four sets of experiments. The first set aimed at evaluating the proposed methods with optimized hyperparameters on state-of-the-art model architectures. The second series evaluating how the accuracy of different models is influenced by non-optimized default hyperparameters. The third set had a more qualitative character to gain an intuition on how the proposed methods influence learning. Lastly, we evaluated how our modifications change the models behavior in a differential privacy setting.

**Default hyperparameters.** All experiments in this series are run with default hyperparameters. To obtain unbiased general hyperparameters, we train and evaluate the MLP on a 2:1 split of the MNIST dataset for ten epochs and a batch size of 1000 using Nevergrad[110] with a budget of 100 (≈3 GPU hours) to determine the learning rate, batch size, the gradient scaling rate $\tau$, the rejuvenation distance $d_{re}$, and the replication factor $\Gamma$. We set the parameters $\tau = 0.09$, $d_{re} = 14$, and $\Gamma = 2$ to obtain the highest accuracy in this setting (cp. supplementary Sec. Predictions with Default Hyperparameters). No other augmentation or regularization was used except for the inherent methods per architecture, i.e., residual connections and batch normalization of the ResNet architecture. We train a network for 100 epochs and retrospectively identify the epoch where accuracy did not increase for five consecutive epochs (early stopping). We report this epoch as a measure of learning speed and report the model's test accuracy at this epoch.

To study how the proposed methods affect the performance of artificial neural architectures, we perform triple cross-validated experiments on the MNIST[55], CIFAR10, and CIFAR100[56] benchmarks. The cross-validation was performed by concatenating the training and the test samples and splitting them into three equal parts, testing one part at a time for three consecutive runs, and averaging the results. We normalize the data samples using the mean and standard deviation calculated on the training splits. We also investigate three network architectures: a shallow learning MLP, a modified version of AlexNet[6], and a ResNet20/32/56[111]. We minimize a cross-entropy loss function[112] using SGD with a learning rate of $\eta = 0.01$ over all classification experiments. The MLP consists of one hidden layer of 1000 neurons for MNIST training and two hidden layers of 3000 neurons each for CIFAR10 and CIFAR100 training. The models with weight splitting have the same number of trainable parameters as the models without weight splitting. Accordingly, we duplicated the activations of each layer to maintain the number of activations of each layer. All experiments together resulted in 1200 h of training time on Nvidia 2080 Ti GPUs. The learning speed is determined in an early stopping scheme. The training machines utilized 10 GPU together with 40 Intel(R) Xeon(R) Silver 4114 CPUs @ 2.20GHz with a total memory of 386G GB on python v3.7 torch v1.5 and torchvision v0.8.

**Qualitative comparison of the loss landscape.** The first experiments evaluate how the proposed biologically meaningful modifications influence the loss landscape of trained models (cp. Fig. 4). We evaluate the shape of the loss landscape and the eigenvalue MinMax, i.e. the ratio of the models Hessian. We used the method and code provided by[62] to visualize the loss landscape on an unmodified and with biomod ResNet18[111]. The hyperparameters were chosen as proposed by[107] and we minimized a cross-entropy loss function[112]. We used the cosine annealing learning rate schedule[113] for 400 epochs with a start learning rate of 0.1, label smoothing of 0.1, weight decay of $5e - 4$, batch size of 256, and the adaptive SAM optimizer[65] with SAM $\rho$ of 1.0 and momentum of 0.9. We also used random cropping of $32 \times 32$ pixels with 4 pixel padding, random horizontal flipping, and channel normalization.

**Resilience to gradient inversion attacks.** We also observed whether biologically motivated gradient noise, as suggested by ref. 114, or splitting the gradient over multiple weights, as suggested by ref. 115, can harden neural networks against gradient inversion attacks[116]. In this experiment, we trained for 100 epochs using SGD with a learning rate of 0.1 and an impulse of 0.9. We normalize the data samples using the mean and standard deviation calculated on the training splits. We use a shallow learning MLP, a modified version of AlexNet[6], and ResNet20 and ResNet32[111], each model trained and untrained. We tested the untrained models because they have stronger gradients and facilitate gradient inversion attacks. The reason is that attackers could attack

during training and start collecting data samples from the training data.

**Tasks and models with optimized hyperparameters.** This study is performed on common benchmark datasets, i.e., CIFAR10 + CIFAR100[56], https://github.com/fastai/imagenette#image, and Tiny ImageNet[117]. The CIFAR benchmarks consist of 10 and 100 classes respectively. Imagewoof consists of 10 similar dog breeds extracted from ImageNet[118]. Tiny ImageNet consists of 200 classes extracted from ImageNet. Each class is represented by 6,000 (CIFAR10), 600 (CIFAR100), 700 (Imagewoof), and 500 (Tiny ImageNet) images. The test set consists of 10,000 images for all benchmarks except for Imagewoof, which consists of 300. The study was conducted on five different random seeds, resulting in a standard deviation smaller 0.1%, which is observed for the BioEfficientNet V2M on CIFAR 100. The experiments are conducted on four ANN architectures: ResNext29 $32 \times 4d$[119], WideResNet $28 \times 10$[120], EfficientNet V2M[121], and SEResNeXt[122] and a transformer architecture SwinTrans V2t[123]. The models were initialized with 0 bias.

We did not use weight splitting for the experiments performed in the tuned setting since we did not observe a positive effect on the tuned regularization. The remaining hyperparameters of the proposed method (gradient scaling rate $\tau \in \{0.00001, 1.0\}$, rejuvenation distance $d_{re} = 0.30$) are optimized by Bayesian optimization[124,125] using NGopt, a method proposed by ref. 126 parallelized with the Asynchronous Successive Halving Algorithm[127,128] with a budget of 400 in about 120 GPU hours. We found the hyperparameters $d_{re} = 6$ and $\tau = 0.5$ to be optimal over the entire series. ResNeXt[119] and WResNet[120] are optimized for image classification with widths from 32–70 pixels. In comparison, EfficientNetV2[121] and SEResNeXt[122] are designed for widths from 224–320 pixels. All architectures are initialized with their PyTorch standard initialization procedure and zero bias.

We perform time series analyses on two benchmarks: the relatively simple Thomas time series[129] is a relatively small dataset with three dimensions, and the Lorenz'96 dataset[130,131] consisting of 396 slow and fast oscillating dimensions derived from ordinary differential equations. We utilize LSTM[132], GRU[133], and FDN[15] for time series prediction as well as a transformer architecture[134]. LSTM and GRU use 512 neurons in a layer with attention. The FDN uses a layer with 225 convolutional kernels with 125 channels. We also used the cosine annealing learning rate schedule[113] for 400 epochs with a start learning rate of 0.1, label smoothing of 0.1, weight decay of $5e − 4$, batch size of 256, adaptive SAM optimizer[65], and optimizer parameters $\rho = 1.0$ and momentum $= 0.9$. To evaluate the time series prediction task, we used the experimental design of[135] predicting time series generated by ordinary differential equations, that is, the single three-dimensional scale Thomas system and the multiscale 396-dimensional Lorenz system. We took 10,000 samples from both systems. We sampled from the Thomas system with time steps of size $dt = 0.0002882$ and parameters $a = 1.85\, b = 10$ with a Lyapunov exponent[136] $LLE \approx 0.76$[137]. The multiscale Lorenz system is sampled with the time step $dt = 0.1$ and the parameters $J = 10$, $b = 10$, $c = 10$, and $h = 1$, with 36 $x$-dimensions and 360 $y$-dimensions; for this parameterization, the Lyapunov exponent $LLE \approx 20$[138].

### Reporting summary
Further information on research design is available in the Nature Portfolio Reporting Summary linked to this article.

## Data availability
The data we used is publicly available. MNIST is available at[55], CIFAR10 and 100 is published from[56], ImageWoof is located here https://github.com/fastai/imagenette#image and tiny Imagenet here[117]. The pretrained Models can be downloaded on this https://docs.pytorch.org/vision/main/models.html. The chaotic time seriessearies thomas[129] and lorenz96[130] are available at https://doi.org/10.6084/m9.figshare.19114151.

## Code availability
To ensure long-term usability and integration with existing PyTorch models, we adhere to software development best practices. These include continuous integration with upstream PyTorch, regular releases on PyPI under the name https://pypi.org/project/pytorch-bio-transformations/, and comprehensive documentation available at https://ceades.github.io/pytorch_bio_transformations/index.html. Our core implementation is accessible at https://github.com/CeadeS/pytorch_bio_transformations, with a reproduction package at https://github.com/CeadeS/BioLearn. These practices aim to create a robust, sustainable tool for the research community." Our reproduction package contains all the code needed to reproduce the reported experiments, including the hyperparameters and network implementations. TheAll code is publicly available at https://doi.org/10.6084/m9.figshare.19114151.

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

## Acknowledgements

This work was supported by funds of the Federal Ministry of Food and Agriculture (BMEL) based on a decision of the parliament of the Federal Republic of Germany via the Federal Office for Agriculture and Food (BLE) under the Federal Programme for Ecological Farming and Other Forms of Sustainable Agriculture (grant: 2819NA106, M.H. and P.M.), by funds of the German Ministry of Education and Research (BMBF) (grant: 01IS20062, M.H. and P.M.), and by funds of the Carl Zeiss Stiftung (grant: P2022-08-006, P.M.). This work was supported by the German Research Foundation (Deutsche Forschungsgemeinschaft, DFG) through grants TE 1172/7-1, SFB1286 subprojects C01, Z01, and the Bundesministerium für Bildung und Forschung (BMBF), grant number: FKZ 01 IS 22 093 A-E.

## Author contributions

M.H. conceived the study, conceptualized the biological mechanisms and their artificial neural network counterparts, designed the methodology and experiments, performed the implementation and evaluation, and wrote the original manuscript draft. M.F.P.B. provided expertise on biological mechanisms, contributing to discussions on bio-ANN differences and co-authoring the related work section. C.T. provided guidance on the biological/ANN designs and evaluation, and contributed to manuscript revisions. P.M. significantly revised the manuscript for structure, tone, and style. All authors reviewed and approved the final manuscript.

## Funding

## Competing interests

The authors declare no competing interests.
