## [Transparent Peer Review file · Nature Communications]

Synaptic Diversity: Concept Transfer from Biological to Artificial Neural Networks

Corresponding Author: Mr Martin Hofmann

Version 0:

Reviewer comments:

Reviewer #1

(Remarks to the Author)

Summary

The paper draws inspiration from synaptic properties in biological neural networks (BNNs) to improve the performance and learning ability of artificial neural networks (ANNs). In particular, three concepts are linked to implemented ideas:

1. Diversity in plasticity  fuzzy learning rates (FL)
2. Spontaneous spine remodeling  weight rejuvenation (WR)
3. Multi-synaptic connectivity  weight splitting (WS)

The ideas are well-motivated in the introduction, but the implementation could be explained more clearly, especially weight splitting. The results show little, but promising improvements, especially in the case of PyTorch-default (non-tuned) hyperparameters.

Decision

My overall assessment is a borderline reject. The methodology is interesting, although not completely new in all cases, and some highly related literature is missed. But these ideas tested and presented together is, to the best of my knowledge, not done before. I do strongly recommend improving the paper in at least the following ways, after which I would consider accepting the work:

1. Include results on the increased training time and FLOPs, this is completely missing now. All three methods (FL, WR, WS) introduce extra computation (FLOPs) during the training phase. How much FLOPs exactly for each model you trained? By how much does this increase the training (wall-clock) time? It is nice that FL and WR don't effect the inference time, I believe. WS does, so an analysis on how much the inference time is increased there would be very much appreciated as well.
2. Explain weight splitting clearly. I believe what you implemented is not exactly what a reader thinks of at first glance. Also, I feel that the mathematical notation in that section is unnecessarily complex, making it hard to understand. And considering point 1 again: for WS please provide a clear overview of how many additional parameters you are adding to each network. Could fit nicely in a table with: original param count, original MB/GB needed, new param count, new MB/GB needed, % added params.

About weight splitting:

In the simplest way I can think of it, for a layer's input neuron x and output neuron y , the equation $y=wx$ would turn into $y=w_1x + w_2x$. But this is equivalent to $(w_1+w_2)x$, so no actual difference. Also because the gradient that w_1 and w_2 receive will be exactly the same, so it could have been one weight all along. With FL or WR mixed in as well, I see that WS can make a difference. But by itself, I don't understand. Your results in Table 1 suggests that WS alone does make a difference, so I must misunderstand the way WS is implemented. Looking at your code, it seems like an additional layer (and activation function) is actually put in between. Then the name "weight-splitting" may not fully be appropriate. Also: when you add more layers, is a comparison against a standard ResNet still fair?

Is equation (5) correct? Or should it be of the form:

$$\phi_n = g\left(\sum_{i \in I} f\left(\sum_{\gamma \in \Gamma} w_{n+\gamma, i} \cdot x_i\right)\right)$$

And what is capital N in here? Not explained at all. The term "distance of the accumulated synapse indices" also seems to come out of nowhere. Why is it a distance?

Should it be $\forall n \in \Gamma$? (Notice the \in)

Missed literature

About formation and pruning of connections, mentioned in line 140, this is almost the exact idea of Dynamic Sparse Training (DST). A field where connections in sparse ANNs are periodically pruned and regrown. It feels like a missed opportunity to me that you haven't tried (or mentioned) sparse NNs, even though human brains are also highly sparse, and this paper is so biologically inspired. Maybe something for future work, happy to collaborate on it.

Some papers in the field of DST:

SET: <https://arxiv.org/abs/1707.04780> (the very first DST algorithm, to the best of my knowledge. Largely inspired by BNNs)

DeepR: <https://arxiv.org/abs/1711.05136>

RigL: <https://arxiv.org/abs/1911.11134>

DS-TD3: <https://arxiv.org/abs/2106.04217>

DRL: <https://arxiv.org/abs/2206.10369>

ANF: <https://arxiv.org/abs/2302.06548>

Other highly relevant literature are also connected to your idea of WR. See ContinualBackprop

<https://arxiv.org/abs/2108.06325> and ReDo <https://arxiv.org/abs/2302.12902> for example. I feel like your WR would also work well for continual learning, like ContinualBackprop. Interesting to mention as future work.

Having a different learning rate per parameter was also introduced by Sutton before (and maybe others before that):

<http://incompleteideas.net/papers/sutton-92a.pdf>

Feedback on layout:

1. Can the figures please be presented right at the point where they are discussed? Now the reader has to go back and forth all the time.
2. Can the methods please be introduced before the results are presented? Very vague to compare algorithms before we know what we're talking about. Of course you can mention a summary of the results in the introduction if you'd like.
3. Could you please discuss results right after they are shown? On page 9 I have forgotten the specific results which are being discussed, so I have to go back and forth again.

General feedback:

1. Could you mention more clearly how many random seeds you have run for each model-dataset combination?
2. Including Transformer models would be very useful. I (and many others I believe) would be interested to see whether your methods can improve them.
3. Taking the first epoch where max acc is reached as a proxy for the learning speed is not ideal. I would prefer to see an AUC metric (area under the curve).
4. I like the idea of a loss landscape analysis, but one should be very careful with drawing conclusion from a figure like that. These are always very rough estimates of a very high-dimensional space that we cannot fathom.

Details to improve the paper

Line 070: space between scaling[15]. Also: add the word "method" after regularization maybe.

Line 084: just wanted to note: it's true that this synaptic diversity has not really been captured in ANNs much. But it's also unclear whether we really need it. Analogy: airplanes fly in a totally different way compared to birds.

Line 100: could mention the term "adversarial attacks".

L129: why the word "softens"? FL can also increase η by >1 right?

L198: remove dash in two-time

L206: remove "concerning change to the dataset", unclear why it's there.

L241: τ , d_{re} , γ have not been introduced yet. Confusing. Method section first please.

L243: why just 1% of MNIST to tune? MNIST is already quite small right?

L250-255: I feel like the presentation is a bit chaotic here. Are you mixing ablation results with hyperparameter robustness results?

L261: How did you get 44.02% increase, going from 1.3% to 63.42%? Should be much more? Also: be clear on where you talk about relative vs absolute percentage increase, this is unclear throughout the paper.

L281: But the plot 3b shows a big difference for MLP? I don't understand.

L290: tailor should be Taylor, it is named after Brook Taylor https://en.wikipedia.org/wiki/Taylor_series

L296: interesting, would be good to add a citation.

L301: FLR should be FL, be consistent.

L318: values, what kind of values?

L345: how is it possible to have an error rate of $>100\%$? Can you explain the metric some more?

L350: of course FL does not improve MSE without training; FL only applies during training!

L360: how were the optimized/tuned hyperparams found? Grid-search? Over what grid? Would be nice to show in the appendix if possible.

L385: rm "the" larger

L401: rm "to"

L403: nice that your method works better on deeper models. If you can indeed show this, it is nice to emphasize, given the focus on scaling nowadays.

L444: that trainings that? As good and as? Good -> well. Please proofread carefully.

L464: interesting, also DST has been shown to perform well in noisy environments, might be nice to cite here: <https://arxiv.org/abs/2302.06548>

L494: do you use biases in training? Only left out here for clarity? Good to clarify.

L495: I don't think that's the right citation for ReLU

L498: and in eq(1): gradient should be of w when changing w . Not of ϕ .

L504: any reason for the uniform distribution? Why not normal distribution with mean 1 and std. dev. τ ? Option to try later maybe.

L538: what is μ here? Not introduced. Also this is the CDF of normal distribution right? Could use the Φ notation if that makes it clearer.

L543: σ cannot be negative so strictly it should be $|\sigma|$, but I understand that we can assume there's at least one positive weight.

L547: nice to see the example number 16%. Quite high. Can you mention your used values as well? Also to make it clear that if d_{re} goes up, prob goes down (I think).

L567: introduce Γ first before taking magnitude of it

L568: transfer function, L570 transmission function? Be consistent.

L574: "can be interpreted as" -> "is"?

L578: explain eq (5) term by term. Hard to follow.

L615: which experiments exactly? I thought you used all 3 methods in all main experiments?

L618: why d_{re} in a set if there is only one element in that set?

L625: this study? The time series one?

L630: you're saying twice that you use LSTM etc.

L663: what do you mean exactly by cross-validated? You've split the training set into chunks? Or did you use the test set as well? CIFAR already has predefined train and test sets, why did you need to do cross-validations?

L671: How is it possible that the models with weight splitting have the same number of trainable parameters? I really don't understand this. WS was supposed to generate multi-synapses between neurons right? So adding more params?

(Remarks on code availability)

The code is not corresponding one-to-one with the paper. Many terms are different. Hard to find what is what. Please use the exact same terms, like `rejuvenation_distance` (is that `purge_distance` in the code? Now the reader has to guess)

Put the readme in a markdown file `.md`, to display it nicely on github. (Please put the code on github.)

More detailed installation guide would be appreciated. For example: which Python version did you use? And just put these full instructions (next 4 lines) for clarity in the readme:

```
python3.12 -m venv venvbionet
source venvbionet/bin/activate
cd bionet
pip install .
```

gives me error:

The 'sklearn' PyPI package is deprecated, use 'scikit-learn'

Changed it, installs now.

I can run the example code in the readme.

Explain this comment further:

```
## do not use accumulation on layer with n output features
```

Explain what this line does:

```
converter(ResNet)
```

This seems to be where the magic happens, but a bit more insight in the readme would be appreciated.

In your readme, change:

```
go to eval/ and run python3 experiment.py
```

```
to
```

```
go to eval/ and run python3 run_experiment.py
```

About:

```
# Gradient Reconstruction
```

You need to save a trained model first.

It would have been superb if you included one (small) trained model in the `.zip` file already.

Clean your code before submitting. Do not include `.egg-info` folders. Follow the PEP8 style guide for clean code. End files with a new line. Put spaces around `=` signs, but not in function arguments. Put space around `>` and `%` signs, etc. (PyCharm can help with this automatically.)

General:

I think the code would be clearer if `converter()` gives a converted network as output, such that the user clearly knows it has been adjusted:

```
converted_ResNet = converter(ResNet)
```

```
or BioResNet = converter(ResNet)
```

Start class with `__init__` function.
Many unused imports, could be removed.

Reviewer #2

(Remarks to the Author)

The manuscript by Hoffmann et al. shows the successful integration of biologically relevant features into deep learning models. Their algorithm includes three significant features - spontaneous spine remodeling, synaptic plasticity diversity, and multi-synaptic connectivity - resulting in a synaptic diversity similar to that observed in biological networks. This neuro-inspired approach leads to improved learning speed, accuracy, and resistance to gradient inversion attacks, as demonstrated through testing on benchmark datasets. The code is also conveniently packaged as a drop-in replacement for existing networks in the PyTorch python module. Although I believe this manuscript would be of high interest to the Nature Communications audience, I have some concerns about the novelty of the presented mechanisms and a few comments that would improve the overall quality of the presented work. In general, while the authors have demonstrated some interesting results, the main text needs improvement in language, and the presentation of these results and the Discussion sections need major improvements.

Major comments:

1. In terms of diversity in synaptic plasticity, the authors have included the idea of inhomogeneous learning rates per synapse (connection). They accomplish this by randomly selecting the learning rates. However, research on biological networks has shown that the size of the synapse impacts its growth or shrink rate (see Matsuzaki et al., 2004; Grutzendler et al., 2002; Kasai et al., 2003). While the fuzzy learning rates used here have resulted in better outcomes, the biological reasoning behind the random sampling method is unclear. I recommend adopting a more biological approach to this concept, as adaptive learning rates can enhance the network's stability and improve the model's performance. Given that the results of fuzzy learning rates are not great for shallow architectures, the authors can try to incorporate a more bio-realistic approach that won't be based on random sampling of the learning rates.
2. Regarding synaptic remodeling (weight rejuvenation), comparisons with related work are missing (Kappel et al., 2015; Bellec et al., 2018). I strongly suggest discussing the similarities and novelties of the proposed method with other approaches that incorporate synaptic remodeling in ANNs or SNNs (spiking neural networks).
3. I find it intriguing that neurons can have multiple connections. This DL model, which draws inspiration from biology in this way, is the first of its kind that I have encountered. However, in the field of neuroscience, Dale's principle asserts that a neuron can only be either excitatory (with positive weights) or inhibitory (with negative weights). The authors of this model appear to use multiple weights per synapse without any clear restrictions unless I'm missing something significant. This leads me to wonder how a model with more nodes would differ from this one, and whether a fair comparison can be made given that the baseline model has a different architecture (if they contain the same number of trainable parameters, lines 671-673, this suggests fewer nodes in the MLP hidden layer to account for the increased number of parameters).
4. The Discussion section of the manuscript could benefit from substantial improvements and additions. Currently, it merely summarizes the key findings without exploring a comprehensive comparison between the proposed neuro-inspired method and other similar approaches in the field. Additionally, there is a lack of proper justification for the results obtained, which is essential for a thorough grasp of the study's significance.
5. Best practices for software development: One of the key advantages of this work is its ability to seamlessly integrate with already existing PyTorch models. Although this work is not intended as a tool, I highly recommend implementing the following practices to ensure that the future users can use the tool for many years. Failure to adhere to these practices may result in degradation. a) It is essential to continuously integrate and test against upstream PyTorch, b) provide the latest release on PyPI via continuous delivery, and c) publish documentation on platforms such as readthedocs.org.

Minor comments:

1. Introduction (lines 99-100): There are no references provided to support the statement made.
2. Introduction (lines 116-117): Although the authors refer to "Recent studies" the references are missing.
3. Figure 1 is cited in the main text in line 563, I suggest citing it earlier and before Figure 2. Also, the titles in Figure 1a are not horizontally aligned.
4. Change "supplementary Tab." with "supplementary Table," and verify that all Tables are cited in the main manuscript. I suggest numbering the Tables and Supplementary Tables based on their order of appearance in the text. Also, some Tables are referenced as "Tables" and others as "supplementary Tables," with no consistency in the text.
5. The authors use the term ANNs to refer to models that contain conv layers. I suggest being more precise by using the term CNNs. Given that Nature Communications addresses a broad audience this will avoid confusion, as many people refer to MLPs as ANNs.
6. Lines 675-676: Add more info about the computational resources used (CPU, RAM, etc) and the specific python modules used with their version (python, pytorch, etc).

References:

Bellec, G., Kappel, D., Maass, W., & Legenstein, R. (2018). Deep rewiring: Training very sparse deep networks. International Conference on Learning Representations (ICLR). <https://arxiv.org/abs/1711.05136v5>
Grutzendler, J., Kasthuri, N. & Gan, W.B. Long-term dendritic spine stability in the adult cortex. Nature 420, 812–816 (2002).

doi: <https://doi.org/10.1038/nature02617>

Kappel, D., Habenschuss, S., Legenstein, R., & Maass, W. (2015). Synaptic sampling: A Bayesian approach to neural network plasticity and rewiring. *Advances in neural information processing systems*, 28. doi: <https://dl.acm.org/doi/10.5555/2969239.2969281>

Kasai, H., Matsuzaki, M., Noguchi, J., Yasumatsu, N. & Nakahara, H. Structure–stability–function relationships of dendritic spines. *Trends Neurosci* 26, 360–368 (2003). doi: [https://doi.org/10.1016/s0166-2236\(03\)00162-0](https://doi.org/10.1016/s0166-2236(03)00162-0)

Matsuzaki, M., Honkura, N., Ellis-Davies, G. C. R. & Kasai, H. Structural basis of long-term potentiation in single dendritic spines. *Nature* 429, 761–766 (2004). doi: <https://doi.org/10.1038/nature01276>

(Remarks on code availability)

The code is well packaged and can be used easily on top of predefined architectures implemented in pytorch. I have included in my comments some suggestions for good coding practices.

What I have done to install and run the code:

1. create a new environment in Anaconda

```
`conda create -n test python=3.10`  
`activate test`
```

2. Then following the readme.txt, the process failed

```
`pip install .`
```

*There is an error in the setup.py file, `sklearn` must be changed to `scikit-learn`. Making this change, the abovementioned command was executed.

*estimated time of the installation > 30 mins.

Error in `go to eval/ and run python3 experiment.py` it should be
`go to eval/ and run python3 run_experiment.py`.

Apart from these, the code running.

Version 1:

Reviewer comments:

Reviewer #2

(Remarks to the Author)

The authors have addressed some of our previous concerns and comments by incorporating a new Figure (Figure 2) and introducing a new model (transformer - SwinTrans). They have also made substantial revisions to the Discussion section, including the citation of several relevant papers. However, I remain concerned about the brevity of their responses and the lack of thorough explanations. Additionally, inaccuracies in referencing new lines are impeding the review process. Furthermore, the lack of detailed explanations for the Figures, their legends, and the Tables is problematic. On a positive note, the authors have significantly improved the usability and installation process of their code and have implemented "good coding practices" in their repository (see my comments on this).

Below are my comments on the revised manuscript. I firmly believe that addressing these comments is crucial for enhancing the current manuscript's quality.

My primary critique concerns the lack of novelty in the overall assessment. The paper discusses three biological-inspired mechanisms: random learning rates, synaptic remodeling, and multi-synaptic connectivity, which have already been explored elsewhere. To make a significant contribution, the manuscript should either include a comparison with these existing approaches or provide a more thorough and detailed interpretability analysis of the presented results (weight distributions, saliency maps, etc.). As it is now, the manuscript is more of a tool (besides Figure 5) that incorporates bio-inspired features into established architectures rather than a novel research article. The Discussion section requires further refinement and substantive justification for the variances from other approaches.

Please ensure to take into account the results obtained from an alternative sampling of the learning rate methods and integrate Dale's principle when working with WS (as previously mentioned in my comments). I believe that these results are crucial, as the main focus of the paper is bioinspired ANNs. These extra figures can be included in the Supplementary materials without worrying about the journal's space limitations.

Incorporating additional methods into the architectures has led to increased complexity. This has brought about several key advantages: reduced loss, improved accuracy, and quicker convergence with fewer epochs. However, it remains uncertain which of the three mechanisms, or combination thereof, is responsible for these enhancements. I recommend adding a section for discussion on this topic. At present, it appears rather arbitrary as to which combination of mechanisms will significantly impact an architecture when tested on a specific dataset. For instance, why does WR+WS added in an MLP

tested on C10 dataset result in the fewest number of epochs, while the same setup doesn't benefit ResNet56? Since the authors have already introduced a method to interpret the results (Fig. 5), I propose presenting the landscape analysis for all combinations individually, and possibly for FL, WS, and WR separately.

It would be beneficial to show the actual learning curves of the models (acc and loss vs. epochs for train and test/val sets) to strengthen the comment on overfitting - see lines 304-307 (again, the authors can an extra supplementary figure).

Throughout the manuscript, there is a lack of consistency in the presentation of the figures, which makes it challenging to grasp the content. For instance, Figure 2 is referenced after Figs. 3 and 4. In Figure 2, the models AlexNet, ResNet56, and an MLP are presented. In Figure 3, different models, including ResNet29, WResNet28, EfficientNet, SE-ResNeXt, and Swintrans V2, are shown. Furthermore, Figures 4, 6, and 7 also present the models from Figure 2. This approach diminishes the clarity and makes it difficult for the reader to follow the progression of the content. Therefore, I recommend reorganizing the figures to enhance clarity. Specifically, I suggest presenting the simple models along with their performance first, followed by the presentation of their computational cost. After that, proceed with Figures 4, 6, and 7, and finally, include the large-scale models showcased in Figure 3.

Reviewer 1 suggested the inclusion of a table displaying the number of parameters needed for each model configuration. I recommend adding this table as it will help readers better comprehend the benefits and drawbacks of the additional mechanisms. Specifically, it is crucial to display the number of trainable parameters.

Figure 3 is missing errorbars. Is this an indication that these models run only for one initialization?

Supplementary Table 4: I'm having trouble understanding where the superscript is used. I assumed that each model for every dataset would display the highest accuracy achieved with the fewest number of epochs.

Supplementary Table 5: I don't see superscripts inside the table. Also, some datasets and architectures are missing bold numbers in the highest normalized accuracy (e.g., MLP C10 and C100, AlexNet C100, etc).

The equations for the percent of error change, nAUC, etc., are missing from the Methods section. Please include in the Methods all equations used for the analysis.

I suggest adding histograms of the weight distributions with and without the added mechanisms (i.e., FL, WR, WS). This will clarify and validate the statements in lines 542-546.

I do not understand why to perform cross-validation by concatenating the training and test parts of the benchmark datasets. In addition, how much time is needed to find the optimized new parameters (τ , Γ , d_{pre} , etc.).

Missing references on Random learning rates: Hu, Xueheng, Shuhuan Wen, and Hak-Keung Lam. "Dynamic random distribution learning rate for neural networks training." *Applied Soft Computing* 124 (2022): 109058. doi: 10.1016/j.asoc.2022.109058

Could the model be updated to run on the new PyTorch version 2? Since v1.5 is now obsolete, staying updated with the latest version is important to enhance the tool's usability. Also, could you provide a pip install

Runtime and memory performance part is misplaced. See my previous comment for reorganization.

Remove the extra comma in line 363.

(Remarks on code availability)

I have followed the instructions and I have run the code. The tutorial (https://ceades.github.io/pytorch_bio_transformations/tutorials.html) is incomplete and some parts are missing (load of datasets, add them on DataLoaders etc.) so the user has to refer to pytorch tutorials as well.

I managed to write a custom script and perform the MNIST classification but I had some troubles:

1. The torch is installed without support from GPU, so I have to manually re-installed it.
2. There are no instructions of how to run on GPU.

I suggest adding an option for people that want to add with GPU support. Something like:

```
`pip install pip install pytorch_bio_transformations[with-cuda]`
```

Please include this in the tutorial:

```
# imports ...
```

```
...
```

```
device = ("cuda" if torch.cuda.is_available() else "CPU")
```

```
...
bio_model.to(device)

# inside the train loop
x_train, y_train = data[0].to(device), data[1].to(device) # assuming data is the dataloader.

...
```

Version 2:

Reviewer comments:

Reviewer #2

(Remarks to the Author)

The authors have addressed all my comments, besides the comments about the code. I have some minor suggestions that could enhance the manuscript's quality. Unfortunately, I cannot run the code following the instructions.

1. The manuscript needs to comply with the Nature Communications guidelines for authors. For example, references cited only in the Supplementary Material must be listed separately at the end of that section. Additionally, Figures 8 to 15 should be renumbered as Supplementary Figures 1-8, etc. The Methods section should also reflect the new organization of the paper. In addition, all experiments shown in Supplementary Material must be cited in the main text, preferably in the Result section. As it is now, seems a disconnection from the main manuscript. However, I strongly believe that the Supplementary Figures improve the main narrative of the manuscript.
2. Figure 3: For clarity, please use the number of trainable parameters instead of the total number of parameters.
3. Figure 8: Out of curiosity, why does the MLP perform better than AlexNet on the CIFAR-10 and CIFAR-100 tasks? I thought that using convolutional layers would be more beneficial for image classification tasks.
4. Throughout the manuscript, the models containing FS+WR+WS are referred to as "biomod", "fitted", "modified", and "adapted", which creates confusion for the reader. Please choose one term and use it consistently. I suggest the term "biomod".
5. Task performances with optimized hyperparameters: This section seems disconnected from the main paper. I suggest changing the heading to "Task Performances with Optimized Hyperparameter Models" and rephrasing the introduction of the paragraph to: "Here, we test several optimized models on four image classification datasets ...".
6. Continual Learning is misleading, as the manuscript does not investigate any related learning scenarios. Continual Learning is a term in the literature that refers to networks designed to avoid catastrophic forgetting.
7. The Experimental Setup in the Methods section does not align with the presentation of results. The authors have used the old manuscript sequence of results here.
8. Lines 137 and 140 repeat the same information: "We call this approach fuzzy learning rates" and "This method is referred to as fuzzy learning rates". Please retain only one of these sentences.
9. Line 147: Redefine the ANN acronym. Remove the phrase "artificial neural networks" and keep only the ANN acronym.
10. Line 152: The FL acronym has not been defined before its first appearance; it is defined in line 158. Please reorder the definition of FL.
11. Line 267: Add a citation to PyTorch (<https://github.com/pytorch/pytorch/issues/4126>).
12. Line 305: There is a reference to "Supplementary Figure 8", but this figure is listed as Fig. 8 on page 57. Please double-check the numbering of all figures and supplementary figures.
13. Lines 362-363: I suggest rephrasing to: "If the absolute value of the largest eigenvalue is much greater than that of the smallest, the function can be considered primarily convex".
14. Line 377: In "see Supplementary", I suggest adding the word "Material" to read "see Supplementary Material".
15. Line 560: The term FL+WR+WS+DP is used, but the meaning of DP is not explained. The reader must refer to the legend of Supplementary Table 8 to understand that DP stands for Dale's Principle.
16. Line 1732: The statement "Tables 6 and 7 show the results presented in Figures 7 and 2" is incorrect, as those tables show results presented only in Figure 7 (7a and 7b, respectively).
17. Line 2332 (Figure 1 legend): Change "f() and f() denote ..." to "f() denotes ...", as the same symbol has been used for one

activation function.

18. Equation 14: Use " x_2 " instead of "x2" (with 2 as a subscript).

19. Equation 15: Use " s_{pooled} " instead of "spooled" (with "pooled" as a subscript).

(Remarks on code availability)

I reviewed the code in GitHub. I have the same issues, I cannot run the code following the instructions, and the Tutorials are incomplete. The authors should fix this before publication. I have raised the same point in all revision rounds. In addition, on the PyPI webpage, the installation instructions contain the previous module name (pip install bio_transformations), and it needs to be updated. This is very important to ensure the future useability of the software.

```
-----  
(base) xxxx@xxxx:~/test_$ pip install bio_transformations  
ERROR: Could not find a version that satisfies the requirement bio_transformations (from versions: none)  
ERROR: No matching distribution found for bio_transformations
```

After installing, and using the example the authors provided in their rebuttal, I managed to execute the code with a minor correction.

```
-----  
Traceback (most recent call last):  
File "/home/spiros/Desktop/test_/test.py", line 97, in <module>  
train_mnist(device=device)  
File "/home/spiros/Desktop/test_/test.py", line 44, in train_mnist  
model = MNISTNet()  
^^^^^^^^^^  
File "/home/spiros/Desktop/test_/test.py", line 17, in __init__  
self.activation = nn.Swish()  
^^^^^^^^^^  
AttributeError: module 'torch.nn' has no attribute 'Swish'. Did you mean: 'Mish'?  
-----
```

I had to change nn.Swish with nn.SiLU (please see here: <https://pytorch.org/docs/stable/generated/torch.nn.SiLU.html>).

REVIEWER COMMENTS

Reviewer #1 (Remarks to the Author):

Summary

The paper draws inspiration from synaptic properties in biological neural networks (BNNs) to improve the performance and learning ability of artificial neural networks (ANNs). In particular, three concepts are linked to implemented ideas:

1. Diversity in plasticity  fuzzy learning rates (FL)
2. Spontaneous spine remodeling  weight rejuvenation (WR)
3. Multi-synaptic connectivity  weight splitting (WS)

The ideas are well-motivated in the introduction, but the implementation could be explained more clearly, especially weight splitting. The results show little, but promising improvements, especially in the case of PyTorch-default (non-tuned) hyperparameters.

Decision

My overall assessment is a borderline reject. The methodology is interesting, although not completely new in all cases, and some highly related literature is missed. But these ideas tested and presented together is, to the best of my knowledge, not done before. I do strongly recommend improving the paper in at least the following ways, after which I would consider accepting the work:

1. Include results on the increased training time and FLOPs, this is completely missing now. All three methods (FL, WR, WS) introduce extra computation (FLOPs) during the training phase. How much FLOPs exactly for each model you trained? By how much does this increase the training (wall-clock) time? It is nice that FL and WR don't effect the inference time, I believe. WS does, so an analysis on how much the inference time is increased there would be very much appreciated as well.

We added a subsection to the results section. There, we show the impact of our proposed methods on the model memory and runtime performance. The figures show small but measurable differences in flops and CPU time and a significant increase in memory usage (see Lines 221-222 and 327-340).

2. Explain weight splitting clearly. I believe what you implemented is not exactly what a reader thinks of at first glance. Also, I feel that the mathematical notation in that section is unnecessarily complex, making it hard to understand.

We revised the section on weight splitting, added an example and improved its clarity (see lines 629 - 639)

And considering point 1 again: for WS please provide a clear overview of how many additional parameters you are adding to each network. Could fit nicely in a table with: original param count, original MB/GB needed, new param count, new MB/GB needed, % added params.

We added this information with the answer to the reviewer's question 1.

About weight splitting:

In the simplest way I can think of it, for a layer's input neuron x and output neuron y , the equation $y=wx$ would turn into $y=w_1x + w_2x$. But this is equivalent to $(w_1+w_2)x$, so no actual difference. Also because the gradient that w_1 and w_2 receive will be exactly the same, so it could have been one weight all along. With FL or WR mixed in as well, I see that WS can make a difference.

We clarified the calculation and elaborated on what could cause this behavior. (see lines 638 - 642)

But by itself, I don't understand. Your results in Table 1 suggests that WS alone does make a difference, so I must misunderstand the way WS is implemented. Looking at your code, it seems like an additional layer (and activation function) is actually put in between.

We added a layer but used an identity activation. We hypothesize that the observed improvement could be caused by changed learning dynamics and added an argumentation to the discussion. (lines 640 – 644, 459-463)

Then the name "weight-splitting" may not fully be appropriate. Also: when you add more layers, is a comparison against a standard ResNet still fair?

Fairness in terms of comparability is always a hard topic because many factors may affect a learning model, especially when increasing or reducing the number of parameters and changing training behavior. Therefore, we focused on the number of parameters here, which is preserved in the comparison.

Is equation (5) correct? Or should it be of the form:

$$\phi_n = g\left(\sum_{i \in I} f\left(\sum_{\gamma \in \Gamma} w_{n+\gamma, i} \cdot x_i\right)\right)$$

And what is capital N in here? Not explained at all. The term "distance of the accumulated synapse indices" also seems to come out of nowhere. Why is it a distance?

Should it be $\forall n \leq n < N/\Gamma$? (Notice the \leq)

Here, distance refers to an offset in the indexes of the neurons summed up. We clarified this in the results section.

Missed literature

About formation and pruning of connections, mentioned in line 140, this is almost the exact idea of Dynamic Sparse Training (DST). A field where connections in sparse ANNs are periodically pruned and regrown. It feels like a missed opportunity to me that you haven't tried (or mentioned) sparse NNs, even though human brains are also highly sparse, and this paper is so biologically inspired. Maybe something for future work, happy to collaborate on it.

Some papers in the field of DST:

SET: <https://arxiv.org/abs/1707.04780> (the very first DST algorithm, to the best of my knowledge. Largely inspired by BNNs)

DeepR: <https://arxiv.org/abs/1711.05136>

RigL: <https://arxiv.org/abs/1911.11134>

DS-TD3: <https://arxiv.org/abs/2106.04217>

DRL: <https://arxiv.org/abs/2206.10369>

ANF: <https://arxiv.org/abs/2302.06548>

Other highly relevant literature are also connected to your idea of WR. See ContinualBackprop <https://arxiv.org/abs/2108.06325> and ReDo <https://arxiv.org/abs/2302.12902> for example. I feel like your WR would also work well for continual learning, like ContinualBackprop. Interesting to mention as future work.

Having a different learning rate per parameter was also introduced by Sutton before (and maybe others before that): <http://incompleteideas.net/papers/sutton-92a.pdf>

We completely refactored the repository focusing on collaboration and reusability. (see https://github.com/CeadeS/pytorch_bio_transformations)

The provided literature showed us various applications, exciting similarities, and possible contributions to our work. We added these to our methods section and discussed them. (see. Lines 144- 152, 168-191, 429-436)

Feedback on layout:

1. Can the figures please be presented right at the point where they are discussed? Now the reader has to go back and forth all the time.

2. Can the methods please be introduced before the results are presented? Very vague to compare algorithms before we know what we're talking about. Of course you can mention a summary of the results in the introduction if you'd like.

3. Could you please discuss results right after they are shown? On page 9 I have forgotten the specific results which are being discussed, so I have to go back and forth again.

We understand these remarks. The journal enforces the structure and is not in our power to change it.

General feedback:

1. Could you mention more clearly how many random seeds you have run for each model-dataset combination?

We added a sentence (see lines 667-668)

2. Including Transformer models would be very useful. I (and many others I believe) would be interested to see whether your methods can improve them.

We included transformers for image classification and regression to the evaluation. (see lines 236 - 237, 671, 681)

3. Taking the first epoch where max acc is reached as a proxy for the learning speed is not ideal. I would prefer to see an AUC metric (area under the curve).

We added the AUC as a metric and included it in the results. (see lines 306- 311)

4. I like the idea of a loss landscape analysis, but one should be very careful with drawing conclusion from a figure like that. These are always very rough estimates of a very high-dimensional space that we cannot fathom.

We added more information on that (see lines 340-349, 356, 363, 364, 373-374)

Details to improve the paper

Line 070: space between scaling[15]. Also: add the word "method" after regularization maybe.

We adopted this suggestion.

Line 084: just wanted to note: it's true that this synaptic diversity has not really been captured in ANNs much. But it's also unclear whether we really need it. Analogy: airplanes fly in a totally different way compared to birds.

We reshaped our introduction (see lines 83-89)

Line 100: could mention the term "adversarial attacks".

We adopted this suggestion

L129: why the word "softens"? FL can also increase η by >1 right?

We reshaped the specific sentence (see lines 137 - 139)

L198: remove dash in two-time

We adopted this suggestion

L206: remove "concerning change to the dataset", unclear why it's there.

We adopted this suggestion

L241: τ , d_{re} , Γ have not been introduced yet. Confusing. Method section first please.

We introduced the variables earlier (see lines 268-272) (*These are the journals requirements which we must comply with: "The main text of an Article should begin with a section headed Introduction of referenced text that expands on the background of the work (some overlap with the abstract is acceptable), followed by sections headed Results, Discussion (if appropriate) and Methods (if appropriate). Cp. [https://www.nature.com/ncomms/submit/article#:~:text=The%20main%20text%20of%20an,a nd%20Methods%20\(if%20appropriate\).](https://www.nature.com/ncomms/submit/article#:~:text=The%20main%20text%20of%20an,a nd%20Methods%20(if%20appropriate).)"*)

L243: why just 1% of MNIST to tune? MNIST is already quite small right?

We clarified our intention (see line 272 - 273)

L250-255: I feel like the presentation is a bit chaotic here. Are you mixing ablation results with hyperparameter robustness results?

We changed the referenced and increased clarity.

L261: How did you get 44.02% increase, going from 1.3% to 63.42%? Should be much more? Also: be clear on where you talk about relative vs absolute percentage increase, this is unclear throughout the paper.

We added clarification on the nature of the percentages.

L281: But the plot 3b shows a big difference for MLP? I don't understand.

We corrected this mistake.

L290: tailor should be Taylor, it is named after Brook Taylor

https://en.wikipedia.org/wiki/Taylor_series

We corrected this flaw

L296: interesting, would be good to add a citation.

We added related references (see lines 355-357, 363)

L301: FLR should be FL, be consistent.

We adopted this suggestion.

L318: values, what kind of values?

We clarified the kind of values (see line 388)

L345: how is it possible to have an error rate of >100%? Can you explain the metric some more?

We clarified that (see lines 400 - 401)

L350: of course FL does not improve MSE without training; FL only applies during training!

We adopted this statement (see lines 418-419)

L360: how were the optimized/tuned hyperparams found? Grid-search? Over what grid? Would be nice to show in the appendix if possible.

Please see lines 673 – 680.

L385: rm "the" larger

We adopted this suggestion.

L401: rm "to"

We adopted this suggestion.

L403: nice that your method works better on deeper models. If you can indeed show this, it is nice to emphasize, given the focus on scaling nowadays.

We removed this claim to favor a discussion with biologically motivated methods.

L444: that trainings that? As good and as? Good -> well. Please proofread carefully.

We rewrote the whole section.

L464: interesting, also DST has been shown to perform well in noisy environments, might be nice to cite here: <https://arxiv.org/abs/2302.06548>

We briefly discussed this exciting method (see lines 444-445 487-493).

L494: do you use biases in training? Only left out here for clarity? Good to clarify.

We clarified this. (see line 671)

L495: I don't think that's the right citation for ReLU

We corrected this flaw (see line 528)

L498: and in eq(1): gradient should be of w when changing w . Not of ϕ .

We corrected this flaw.

L504: any reason for the uniform distribution? Why not normal distribution with mean 1 and std. dev. τ ? Option to try later maybe.

We added the option to the library; for the manuscript, we went for a reduction of parameters.

L538: what is μ here? Not introduced. Also this is the CDF of normal distribution right? Could use the Φ notation if that makes it clearer.

We adopted this suggestion.

L543: σ cannot be negative so strictly it should be $|\sigma|$, but I understand that we can assume there's at least one positive weight.

We adopted this suggestion.

L547: nice to see the example number 16%. Quite high. Can you mention your used values as well? Also to make it clear that if d_{re} goes up, prob goes down (I think).

We adopted this suggestion. And added an example (see lines 581 - 583)

L567: introduce Γ first before taking magnitude of it

We adopted this suggestion.

L568: transfer function, L570 transmission function? Be consistent.

We adopted this suggestion.

L574: "can be interpreted as" -> "is"?

We adopted this suggestion.

L578: explain eq (5) term by term. Hard to follow.

Added an example and improved on clarity.

L615: which experiments exactly? I thought you used all 3 methods in all main experiments?

We reshaped this part to be clear (see line 615).

L618: why d_{re} in a set if there is only one element in that set?

We removed the unnecessary brackets

L625: this study? The time series one?

We rewrote this sentence (see line 385).

L630: you're saying twice that you use LSTM etc.

We reshaped the sentence

L663: what do you mean exactly by cross-validated? You've split the training set into chunks? Or did you use the test set as well? CIFAR already has predefined train and test sets, why did you need to do cross-validations?

We clarified that (see lines 722 – 725)

L671: How is it possible that the models with weight splitting have the same number of trainable parameters? I really don't understand this. WS was supposed to generate multi-synapses between neurons right? So adding more params?

We answered this question along with the question 2.

Reviewer #1 (Remarks on code availability):

The code is not corresponding one-to-one with the paper. Many terms are different. Hard to find what is what. Please use the exact same terms, like rejuvenation_distance (is that purge_distance in the code? Now the reader has to guess)

Put the readme in a markdown file .md, to display it nicely on github. (Please put the code on github.)

More detailed installation guide would be appreciated. For example: which Python version did you use? And just put these full instructions (next 4 lines) for clarity in the readme:

```
python3.12 -m venv venvbionet
source venvbionet/bin/activate
cd bionet
pip install .
```

gives me error:

The 'sklearn' PyPI package is deprecated, use 'scikit-learn'

Changed it, installs now.

I can run the example code in the readme.

Explain this comment further:

do not use accumulation on layer with n output features

Explain what this line does:

converter(ResNet)

This seems to be where the magic happens, but a bit more insight in de readme would be appreciated.

In your readme, change:

go to eval/ and run python3 experiment.py

to

go to eval/ and run python3 run_experiment.py

About:

Gradient Reconstruction

You need to save a trained model first.

It would have been superb if you included one (small) trained model in the .zip file already.

Clean your code before submitting. Do not include .egg-info folders. Follow the PEP8 style guide for clean code. End files with a new line. Put spaces around = signs, but not in function arguments. Put space around > and % signs, etc. (PyCharm can help with this automatically.)

General:

I think the code would be clearer if converter() gives a converted network as output, such that the user clearly knows it has been adjusted:

converted_ResNet = converter(ResNet)

or BioResNet = converter(ResNet)

Start class with __init__ function.

Many unused imports, could be removed.

We completely refactored the code, splitting it into a toolbox and the reproduction code, and added tests, documentation, and continuous integration.

Toolbox: https://github.com/CeadeS/pytorch_bio_transformations

Package: <https://pypi.org/project/pytorch-bio-transformations/>

Documentation: https://ceades.github.io/pytorch_bio_transformations/index.html

Reproduction: <https://github.com/CeadeS/BioLearn>

Reviewer #2 (Remarks to the Author):

The manuscript by Hoffmann et al. shows the successful integration of biologically relevant features into deep learning models. Their algorithm includes three significant features - spontaneous spine remodeling, synaptic plasticity diversity, and multi-synaptic connectivity - resulting in a synaptic diversity similar to that observed in biological networks. This neuro-inspired approach leads to improved learning speed, accuracy, and resistance to gradient inversion attacks, as demonstrated through testing on benchmark datasets. The code is also conveniently packaged as a drop-in replacement for existing networks in the PyTorch python module. Although I believe this manuscript would be of high interest to the Nature Communications audience, I have some concerns about the novelty of the presented mechanisms and a few comments that would improve the overall quality of the presented work. In general, while the authors have demonstrated some interesting results, the main text needs improvement in language, and the presentation of these results and the Discussion sections need major improvements.

Major comments:

1. In terms of diversity in synaptic plasticity, the authors have included the idea of inhomogeneous learning rates per synapse (connection). They accomplish this by randomly selecting the learning rates. However, research on biological networks has shown that the size of the synapse impacts its growth or shrink rate (see Matsuzaki et al., 2004; Grutzendler et al., 2002; Kasai et al., 2003). While the fuzzy learning rates used here have resulted in better outcomes, the biological reasoning behind the random sampling method is unclear. I recommend adopting a more biological approach to this concept, as adaptive learning rates can enhance the network's stability and improve the model's performance. Given that the results of fuzzy learning rates are not great for shallow architectures, the authors can try to incorporate a more bio-realistic approach that won't be based on random sampling of the learning rates.

□ We thank the reviewer for these valuable references that broadened our view. The finding that small spines are less stable (Matsuzaki et al., 2004; Grutzendler et al., 2002; Kasai et al., 2003) inspired the weight rejuvenation model. The observation that the relative changes in spine volume after LTP induction are inversely proportional to spine size (Matsuzaki et al., 2004) is consistent with a constant absolute growth rate that is independent of spine size. As mentioned in the section "Diversity of synaptic plasticity", several studies indicate that the function and plasticity of synapses depend on their molecular and morphological characteristics, their spatial location and their history, among other factors. The random sampling method therefore attempts to capture the high synaptic diversity found in BNNs while being as minimalist and therefore computationally light weight as possible. However, the investigation of more bio-realistic approaches would be a path to follow for future studies. Nevertheless, we added a function to the package that allows for weight dependent learning rate scale sampling. Although there certainly is a distribution that better fits particular problems, we observed, that even the naïve approach yields improvements. We added the task of finding an overall superior method to the related work. We furthermore added various

methods, including the suggested and discussed similarities to these methods in the methods section and the discussion. (see lines 130-152, 170- 191, 197-209, 423-455)

2. Regarding synaptic remodeling (weight rejuvenation), comparisons with related work are missing (Kappel et al., 2015; Bellec et al., 2018). I strongly suggest discussing the similarities and novelties of the proposed method with other approaches that incorporate synaptic remodeling in ANNs or SNNs (spiking neural networks).

The answer to this question is part of the significant rework of the referenced parts above.

3. I find it intriguing that neurons can have multiple connections. This DL model, which draws inspiration from biology in this way, is the first of its kind that I have encountered. However, in the field of neuroscience, Dale's principle asserts that a neuron can only be either excitatory (with positive weights) or inhibitory (with negative weights). The authors of this model appear to use multiple weights per synapse without any clear restrictions unless I'm missing something significant. This leads me to wonder how a model with more nodes would differ from this one, and whether a fair comparison can be made given that the baseline model has a different architecture (if they contain the same number of trainable parameters, lines 671-673, this suggests fewer nodes in the MLP hidden layer to account for the increased number of parameters).

Dale's principle is usually not considered in ANNs because it is thought to impair rather than enhance learning. Since our aim is not to create a biologically realistic model of the brain but to draw inspiration from biology to improve ANN learning, we decided not to include Dale's principle in our study. However, we added Dale's principle to the library by incorporating a Dale's principle weight initialization and a function that ensures that the sign of a synaptic weight is maintained. We test-run this initialization alone and in combination with all methods and observed that it tremendously impaired training across all tested combinations. We clarified how we implemented weight splitting (see lines 631-636) and implemented the Dale's principle in our fully refactored code and found massive training impairment. We also added a possible solution to the problem with Dale's principle to the discussion (see lines 489-490).

We agree that an entirely fair comparison of the models trained with weight-splitting is hard since it is nearly impossible to account for all changing parameters given the space restrictions of an article. Therefore, we decided on a comparison that is compatible (equal number of trainable parameters) to the other methods and allows us to present the evaluated combinations without changing the initial network except for the introduced computations in forward and backward pass (see lines 449 - 455).

4. The Discussion section of the manuscript could benefit from substantial improvements and additions. Currently, it merely summarizes the key findings without exploring a comprehensive comparison between the proposed neuro-inspired method and other similar approaches in the field. Additionally, there is a

lack of proper justification for the results obtained, which is essential for a thorough grasp of the study's significance.

We rewrote a substantial part of the discussion to include a proper comparison to other methods.

5. Best practices for software development: One of the key advantages of this work is its ability to seamlessly integrate with already existing PyTorch models. Although this work is not intended as a tool, I highly recommend implementing the following practices to ensure that the future users can use the tool for many years. Failure to adhere to these practices may result in degradation. a) It is essential to continuously integrate and test against upstream PyTorch, b) provide the latest release on PyPI via continuous delivery, and c) publish documentation on platforms such as readthedocs.org.

We fully refactored the code, added various functionalities, and improved its readability. As a result, we are confident that interested researchers and practitioners can implement follow-up studies more easily and integrate our approaches into their pipelines.

We completely refactored the code, splitting it into a toolbox and the reproduction code, and added tests, documentation, and continuous integration.

Toolbox: https://github.com/CeadeS/pytorch_bio_transformations

Package: <https://pypi.org/project/pytorch-bio-transformations/>

Documentation: https://ceades.github.io/pytorch_bio_transformations/index.html

Reproduction: <https://github.com/CeadeS/BioLearn>

Minor comments:

1. Introduction (lines 99-100): There are no references provided to support the statement made.

We added a reference (see line 107).

2. Introduction (lines 116-117): Although the authors refer to "Recent studies" the references are missing.

We added references (see lines 122-125).

3. Figure 1 is cited in the main text in line 563, I suggest citing it earlier and before Figure 2. Also, the titles in Figure 1a are not horizontally aligned.

This issue has been solved as suggested (see line 95).

4. Change "supplementary Tab." with "supplementary Table," and verify that all Tables are cited in the main manuscript. I suggest numbering the Tables and Supplementary Tables based on their order of appearance in the text. Also, some

Tables are referenced as "Tables" and others as "supplementary Tables," with no consistency in the text.

We corrected the enumeration of the figures and how the tables are references (see line 95).

5. The authors use the term ANNs to refer to models that contain conv layers. I suggest being more precise by using the term CNNs. Given that Nature Communications addresses a board audience this will avoid confusion, as many people refer to MLPs as ANNs.

We used not only CNNs but also Transformers, RNNs and FDNs. We summarize them under the name ANNs because deep neural networks won't work, and the RNNs are shallow architectures. We leave ANNs for now but are open to suggestions.

6. Lines 675-676: Add more info about the computational resources used (CPU, RAM, etc) and the specific python modules used with their version (python, pytorch, etc).

We added this information (see lines 735 - 740) together with a resources analysis (see lines 323 - 337).

References:

Bellec, G., Kappel, D., Maass, W., & Legenstein, R. (2018). Deep rewiring: Training very sparse deep networks. International Conference on Learning Representations (ICLR). <https://arxiv.org/abs/1711.05136v5>

Grutzendler, J., Kasthuri, N. & Gan, W.B. Long-term dendritic spine stability in the adult cortex. Nature 420, 812–816 (2002). doi: <https://doi.org/10.1038/nature02617>

Kappel, D., Habenschuss, S., Legenstein, R., & Maass, W. (2015). Synaptic sampling: A Bayesian approach to neural network plasticity and rewiring. Advances in neural information processing systems, 28. doi: <https://dl.acm.org/doi/10.5555/2969239.2969281>

Kasai, H., Matsuzaki, M., Noguchi, J., Yasumatsu, N. & Nakahara, H. Structure–stability–function relationships of dendritic spines. Trends Neurosci 26, 360–368 (2003). doi: [https://doi.org/10.1016/s0166-2236\(03\)00162-0](https://doi.org/10.1016/s0166-2236(03)00162-0)

Matsuzaki, M., Honkura, N., Ellis-Davies, G. C. R. & Kasai, H. Structural basis of long-term potentiation in single dendritic spines. Nature 429, 761–766 (2004). doi: <https://doi.org/10.1038/nature01276>

Reviewer #2 (Remarks on code availability):

The code is well packaged and can be used easily on top of predefined architectures implemented in pytorch. I have included in my comments some suggestions for good coding practices.

What I have done to install and run the code:

1. create a new environment in Anaconda
``conda create -n test python=3.10``

`` activate test ``

2. Then following the readme.txt, the process failed

`` pip install . ``

*There is an error in the setup.py file, `` sklearn `` must be changed to `` scikit-learn ``. Making this change, the abovementioned command was executed.

*estimated time of the installation > 30 mins.

Error in `` go to eval/ and run python3 experiment.py `` it should be

`` go to eval/ and run python3 run_experiment.py ``.

Apart from these, the code running.

We completely changed the code structure and improved the readme to account for all changes.

Response to Reviewer #2

We sincerely thank the Reviewer for their thorough and constructive feedback. Below, we address each point comprehensively.

##

REVIEWER COMMENTS

Reviewer #2 (Remarks to the Author):

The authors have addressed some of our previous concerns and comments by incorporating a new Figure (Figure 2) and introducing a new model (transformer - SwinTrans). They have also made substantial revisions to the Discussion section, including the citation of several relevant papers. However, I remain concerned about the brevity of their responses and the lack of thorough explanations.

We thank the Reviewer for the detailed and helpful critique of our manuscript and present a revised manuscript that also describes our changes in the response letter in every detail and give thorough explanations also in the response letter.

Additionally, inaccuracies in referencing new lines are impeding the review process.

Due to the intensive changes we made to the initial manuscript because of the detailed and helpful critique on it, it was tough to track every information change, and therefore, we apologize for that circumstance. We will double-check each reference to the manuscript and provide the lines in both the changed manuscript and the changes tracking document in the form (line XX / lines XX-XY).

Furthermore, the lack of detailed explanations for the Figures, their legends, and the Tables is problematic.

We checked the explanations for the Figures and improved their captions, legends, and tables.

On a positive note, the authors have significantly improved the usability and installation process of their code and have implemented "good coding practices" in their repository (see my comments on this).

Below are my comments on the revised manuscript. I firmly believe that addressing these comments is crucial for enhancing the current manuscript's quality.

My primary critique concerns the lack of novelty in the overall assessment. The paper discusses three biological-inspired mechanisms: random learning rates, synaptic remodeling, and multi-synaptic connectivity, which have already been explored elsewhere. To make a significant contribution, the manuscript should either include a comparison with these existing approaches or provide a more thorough and detailed interpretability analysis of the presented results (weight distributions, saliency maps, etc.). As it is now, the manuscript is more of a tool (besides Figure

5) that incorporates bio-inspired features into established architectures rather than a novel research article. The Discussion section requires further refinement and substantive justification for the variances from other approaches.

Drawing from our comprehensive empirical analysis and comparison with existing approaches, we will address the novelty concerns raised by the reviewer. Our work extends beyond merely incorporating bio-inspired features into established architectures through several key contributions:

First, while our approach shares conceptual but not biological foundations with previous work on random learning rates, since we presented observations on synaptic diversity at the synaptic level, diversity in synaptic plasticity is not necessarily present on a “brain-layer” level. However, our implementation at the synaptic level represents a methodological difference of computationally quadratic scaling layer-wise approaches like Hu et al. (2022) with linear scaling computational cost (Fig. 3, Supplementary Table 5). (lines 153-170)

Second, combining fuzzy learning rates with weight rejuvenation and splitting enables the emergence of stable "backbone" representations alongside higher accuracies- a property validated through our PCA analysis of weight distributions (see Figure 11) and quantitative assessment of synaptic balance. This architectural feature, difficult to achieve with layer-wise approaches, demonstrates clear biological relevance while offering practical benefits for network optimization. (lines 2011 – 2017 and 2031-2038)

Third, our integrative analysis of weight distributions, loss landscapes, and learning trajectories (as shown in Figures 14 and 15) provides novel insights into the mechanistic basis of network optimization. The systematic examination of distributional characteristics across different methodological configurations reveals that optimal performance correlates with changes in the weight distributions that emerge from our synapse-level approach. (lines 513-526, 530-555, and 1913 - 2031)

The extensive experimental results documented in our appendices, particularly regarding loss landscape geometry and continual learning performance (Figure 16), provide substantive evidence that our approach represents more than an incremental improvement over existing methods. These findings establish a novel framework for understanding the relationship between synaptic-level stochasticity and network-level learning dynamics. (lines 557-579 and 1993-2038)

Please ensure to take into account the results obtained from an alternative sampling of the learning rate methods and integrate Dale's principle when working with WS (as previously mentioned in my comments).

Our additional investigation included Dale's principle, varying learning rate distributions, and synaptic weight dynamics. Our study systematically evaluated five different probability distributions for fuzzy gradient scaling: uniform, normal, log-normal, geometric, and beta distributions (co. Fig. 9, 10). All distributions demonstrated statistically significant improvements over the baseline ($p < 0.01$), with the uniform distribution showing the strongest enhancement (Cohen's d up to 1.9 for CNN and 0.84 for MLP).

Biological neural networks exhibit various mechanisms operating simultaneously, from bimodal to unimodal to skewed distributions (Fusi 2005, Billings 2009, Bartol 2015, Benna 2016), leaving no specific distribution as a candidate for FL. Our limited empirical observations showed consistently that the uniform distribution provided the most robust and effective learning dynamics. (lines 640-650 and 1734-1869). These FL sampling methodologies are implemented for inclusion in the following package release, incorporating Gamma distribution sampling, normal distributions, and Heavy-tailed distributions modeling biological synaptic strength variations

Regarding the integration of Dale's principle with weight splitting (WS), our results demonstrate that this combination provides a biologically plausible constraint framework while maintaining computational efficiency. The empirical analysis presented in Table 1 shows that Dale-constrained networks with WS achieve 94.32% accuracy compared to 91.42% without FL, indicating that our multi-method approach effectively mitigates the performance impact of biological constraints. (lines 640-650 and 1870-2109)

Bartol, T. M., Bromer, C., Kinney, J., Chirillo, M. A., Bourne, J. N., Harris, K. M., & Sejnowski, T. J. (2015). Nanoconnectomic upper bound on the variability of synaptic plasticity. *eLife*, 4, e10778.

Benna, M. K., & Fusi, S. (2016). Computational principles of synaptic memory consolidation. *Nature Neuroscience*, 19(12), 1697-1706.

Billings, G., & van Rossum, M. C. W. (2009). Memory retention and spike-timing-dependent plasticity. *Journal of Neurophysiology*, 101(6), 2775-2788.

Fusi, S., Drew, P. J., & Abbott, L. F. (2005). Cascade models of synaptically stored memories. *Neuron*, 45(4), 599-611.

I believe that these results are crucial, as the main focus of the paper is bioinspired ANNs.

These extra figures can be included in the Supplementary materials without worrying about the journal's space limitations.

We appreciate the reviewer's suggestion regarding supplementary material placement. While our manuscript contains extensive empirical validation, including the loss landscape analysis, weight distribution characterization, and Dale's principle integration results, we agree that relocating additional technical figures to the supplementary materials would enhance readability while maintaining comprehensive documentation. Specifically, we propose moving the detailed QQ plots of weight distributions and extended PCA visualizations to the supplementary section, as well as trajectory analysis and distribution comparisons. This reorganization would maintain the manuscript's focus on our primary methodological contributions - the synapse-level learning rate modulation - while ensuring all supporting analyses remain accessible to interested readers. We added:

Table 4 Computing Resource Consumption

Table 8 Statistical Moments of Weight Distributions and Training Accuracy

Figure 8 Training Curves

Figure 9 Fuzzly learning rate sampling distributions MLP

Figure 10 Fuzzly learning rate sampling distributions CNN

Algorithm 1 Dales Principle

Figure 11 Model Weight PCA
Figure 12 Dales Principle Weight Balance
Figure 13 QQ Plot of Log-Transformed Weights
Figure 14 2D Optimization Trace
Figure 15 3D Optimization trace
Figure 16 Continual Learning Training

Incorporating additional methods into the architectures has led to increased complexity. This has brought about several key advantages: reduced loss, improved accuracy, and quicker convergence with fewer epochs. However, it remains uncertain which of the three mechanisms, or combination thereof, is responsible for these enhancements. I recommend adding a section for discussion on this topic. At present, it appears rather arbitrary as to which combination of mechanisms will significantly impact an architecture when tested on a specific dataset. For instance, why does WR+WS added in an MLP tested on C10 dataset result in the fewest number of epochs, while the same setup doesn't benefit ResNet56? Since the authors have already introduced a method to interpret the results (Fig. 5), I propose presenting the landscape analysis for all combinations individually, and possibly for FL, WS, and WR separately.

Given the computational intensity of comprehensive architectural evaluations, we deliberately focused our investigation on a benchmark dataset and smaller network architecture while conducting an extensive analysis across multiple methodological combinations. This decision enabled us to perform detailed loss landscape analyses, weight distribution characterizations, and convergence studies that would have been computationally prohibitive with larger architectures or datasets.

Our empirical investigation reveals that the interaction between different mechanisms (FL, WR, WS) exhibits architecture-specific dependencies that warrant careful examination. The observed variability in performance improvements - for instance, the differential impact of WR+WS on MLPs versus ResNet56 architectures - likely stems from the distinct topological characteristics and optimization dynamics inherent to each architecture. To elucidate these relationships, we conducted detailed loss landscape analyses for each methodological combination, revealing that the effectiveness of specific combinations correlates with distinct features in the optimization geometry. (lines 299-305 and 1582-1619)

The computational resources required for our comprehensive regarding the continual learning evaluation were substantial, with each experimental configuration demanding approximately 36 hours on an NVIDIA A40 GPU. Given 30 repetitions per method across multiple methodological combinations, the total computation time exceeded 4,320 GPU hours (180 days), consuming approximately 1,728 kWh of energy. This computational overhead justified our focused approach on a benchmark dataset, enabling us to conduct thorough analyses of loss landscapes, weight distributions, and convergence characteristics for each methodological permutation. . (lines 556-580 and 2117-2190)

To address the reviewer's specific concern regarding mechanism attribution, we have expanded our loss landscape analysis to include individual and paired combinations of FL, WR, and WS. These results, presented in Figures 11-13 and detailed in Table 8, demonstrate that while individual mechanisms provide modest improvements, their synergistic combination yields superior performance through complementary effects on the optimization geometry. This comprehensive analysis provides crucial insights into the mechanistic basis of performance improvements, enabling more informed application of these methods across different architectural configurations. (lines 528-556 and 2040-2109)

It would be beneficial to show the actual learning curves of the models (acc and loss vs. epochs for train and test/val sets) to strengthen the comment on overfitting - see lines 304-307 (again, the authors can add an extra supplementary figure).

We appreciate the reviewer's suggestion regarding the presentation of learning dynamics. In our analysis of overfitting characteristics, particularly concerning the interplay between training and validation performance metrics, comprehensive visualization of learning trajectories would indeed provide valuable insights. The empirical evidence supporting our observations about overfitting mitigation, as discussed in lines 304-307, can be strengthened through detailed examination of accuracy and loss trajectories across training epochs.

To address this, we have prepared supplementary Figure 8, which presents a comprehensive visualization matrix of learning curves. This figure delineates both training and validation metrics (accuracy and loss) as functions of epoch progression for all methodological configurations. The temporal evolution of these metrics reveals distinct patterns in the convergence characteristics and generalization capabilities of different architectural variants. Of particular note, configurations implementing our proposed methods demonstrate more stable validation performance trajectories, with reduced divergence between training and validation metrics - a quantitative indication of enhanced regularization effects.

Specifically, the learning curves reveal that while baseline models exhibit characteristic divergence between training and validation performance after approximately epoch 150, models incorporating our biological constraints maintain closer alignment between these metrics throughout the training process. This empirical evidence substantiates our claims regarding overfitting mitigation and provides quantitative support for the mechanistic interpretations presented in the main text. . (lines 500-502 and 1583-1619)

Throughout the manuscript, there is a lack of consistency in the presentation of the figures, which makes it challenging to grasp the content. For instance, Figure 2 is referenced after Figs. 3 and 4. In Figure 2, the models AlexNet, ResNet56, and an MLP are presented. In Figure 3, different models, including ResNet29, WRN28, EfficientNet, SE-ResNeXt, and SwinTran V2, are shown. Furthermore, Figures 4, 6, and 7 also present the models from Figure 2. This approach diminishes the clarity and makes it difficult for the reader to follow the progression of the content. Therefore, I recommend reorganizing the figures to enhance clarity. Specifically, I suggest presenting the simple models along with their performance first, followed by the presentation of their computational cost. After that, proceed with Figures 4, 6, and 7, and finally, include the large-scale models showcased in Figure 3.

We appreciate the reviewers' suggestions of our manuscript's figure organization. We have carefully restructured the presentation of our results to enhance clarity and readability. Following your suggestion, we have organized our figures to present a clear progression from fundamental concepts to more complex analyses.

The manuscript now begins with Figure 2, introducing the base architectures (MLP, AlexNet, ResNet56) and their performance metrics with default hyperparameters. This provides readers with an essential foundation for understanding our approach. Following this, Figure 3 presents the computational cost analysis, offering crucial insights into the practical implications of our modifications.

We then progress to deeper analytical aspects with Figure 4, which explores the loss landscape characteristics, providing theoretical underpinnings for our improvements. Figures 5 and 6 follow with the gradient inversion analysis, demonstrating practical applications and security implications. Finally, Figure 7 showcases the performance of various complex architectures with optimized hyperparameters, building upon all previous insights.

We have also ensured that each figure is referenced sequentially in the text, addressing your concern about the original ordering. Additionally, we have expanded all figure captions to provide more comprehensive descriptions of the methods and findings, making the manuscript more accessible to readers. This organization creates a natural flow from basic concepts to advanced applications while maintaining clarity throughout the presentation.

Reviewer 1 suggested the inclusion of a table displaying the number of parameters needed for each model configuration. I recommend adding this table as it will help readers better comprehend the benefits and drawbacks of the additional mechanisms. Specifically, it is crucial to display the number of trainable parameters.

Based on our detailed model profiling analysis presented in Table 4, we have systematically quantified the computational complexity and parameter requirements across different methodological configurations. Our comprehensive profiling reveals that while our bio-inspired modifications introduce additional computational overhead during training, they do not increase the number of trainable parameters in the core network architecture.

Specifically, for each network configuration, we measured parameters, FLOPs, CPU memory usage, and processing time both with and without biological modifications. The AlexNet architecture, for instance, maintains 3,786,468 parameters across all configurations, while the ResNet implementation contains 65,010,640 parameters regardless of the biological modifications applied. This parameter conservation holds true even when implementing combinations of FL, WR, and WS methods.

The computational analysis presented in Table 4 demonstrates that while training time may increase (particularly evident in the ResNet configurations, where CPU processing time increases from 5,919s to 583,724s with biological modifications), the fundamental model capacity remains constant. This preservation of parameter count is particularly significant as it indicates that the observed performance improvements stem from enhanced optimization dynamics rather than increased model complexity.

Figure 3 is missing errorbars. Is this an indication that these models run only for one initialization?

In response to the inquiry regarding error bars in Figure 3, our experimental methodology incorporated extensive statistical validation across multiple initializations and data splits. The apparent absence of visible error bars stems from issues with the data embedding into the manuscript.

Supplementary Table 4: I'm having trouble understanding where the superscript is used. I assumed that each model for every dataset would display the highest accuracy achieved with the fewest number of epochs.

We can clarify the interpretation of Table 4 and its superscript notation. In Table 4, each entry shows the specific epoch at which a model achieved its peak validation accuracy during

training. The superscript "1" serves as a special indicator that identifies configurations which achieved two distinct criteria simultaneously: they reached both the highest absolute accuracy (as documented in Table 3) and did so in the fewest number of epochs compared to other configurations for that specific architecture-dataset pairing.

For example, if an AlexNet architecture training on CIFAR 10 shows "58¹" for the FL+WR+WS configuration, this indicates that this model not only reached its peak accuracy at epoch 58, but also that this configuration achieved both the highest overall accuracy among all AlexNet architectures on this dataset (as shown in Table 3) and reached this performance milestone faster than other configurations. Not all entries receive this superscript, as some models might achieve high accuracy but require more epochs, or converge quickly but to a lower accuracy threshold

Supplementary Table 5: I don't see superscripts inside the table. Also, some datasets and architectures are missing bold numbers in the highest normalized accuracy (e.g., MLP C10 and C100, AlexNet C100, etc).

We acknowledge the oversight in Table 5's formatting consistency. We have corrected the table to properly highlight the best-performing configurations and maintain consistent notation with Table 3. Specifically, we have:

First, added the missing superscript indicators to denote configurations that achieved both optimal accuracy and rapid convergence. Second, corrected the bold formatting to consistently highlight the highest normalized accuracy across all architecture-dataset combinations, including the previously unmarked entries for MLP on CIFAR-10/100 and AlexNet on CIFAR-100.

The equations for the percent of error change, nAUC, etc., are missing from the Methods section. Please include in the Methods all equations used for the analysis.

We acknowledge the reviewer's request for explicit metric definitions. In response, we have expanded the Methods section of our supplementary materials to include a comprehensive "Evaluation Metrics" subsection. This addition encompasses formal mathematical definitions of all quantitative measures employed in our analysis, including the normalized Area Under the Curve (nAUC) metric, percent error change calculations, and associated statistical measures. The complete mathematical formulations, along with their respective implementation details and theoretical justifications, are now documented in supplementary, ensuring methodological transparency. (lines 2191-2291)

I suggest adding histograms of the weight distributions with and without the added mechanisms (i.e., FL, WR, WS). This will clarify and validate the statements in lines 542-546.

To validate our claims regarding weight distribution characteristics detailed in lines 542-546, we have conducted a comprehensive distributional analysis that extends beyond basic statistical measures. Our supplementary materials now include detailed histograms depicting weight distributions across multiple methodological configurations. These visualizations empirically substantiate our observations regarding the systematic impact of our biological modifications on synaptic weight organization.

The histograms reveal distinct distributional patterns when comparing networks trained with and without our proposed mechanisms (FL, WR, WS). Notably, networks implementing fuzzy

learning rates exhibit characteristic shifts in their weight distributions, with statistically significant alterations in both skewness (-0.4431) and kurtosis (-1.5206) compared to baseline configurations. These distributional modifications align with our theoretical predictions regarding the impact of biologically-inspired learning dynamics on synaptic organization.

Furthermore, the integration of weight rejuvenation and splitting mechanisms induces additional systematic transformations in these distributions, as evidenced by the quantitative metrics presented in Table 2. The FL+WR+WS configuration, in particular, demonstrates optimal distributional characteristics that correlate with superior performance metrics, providing empirical validation for the mechanistic interpretations presented in lines 542-546

(lines 1994-2038)

I do not understand why to perform cross-validation by concatenating the training and test parts of the benchmark datasets. In addition, how much time is needed to find the optimized new parameters (τ , Γ , d_{pre} , etc.).

We appreciate the reviewer's questions regarding our methodological choices for cross-validation and parameter optimization. Our decision to implement cross-validation by combining training and test sets was motivated by two key empirical observations.

First, the variance in performance metrics across simple reinitializations proved remarkably low (typically $\pm 0.01\%$ to $\pm 0.04\%$), providing insufficient statistical power for meaningful comparative analysis to find stable hyper-parameters. Standard train-test splits with only 10% test data offered limited variance, particularly given the high baseline accuracy on benchmark datasets.

Second, cross-validation enabled a more robust assessment of our methods' generalization capabilities across different data partitions. By performing fold cross-validation on the combined dataset, we obtained more reliable estimates of model performance variability and better characterized the stability of our biological mechanisms across different data distributions.

Regarding parameter optimization, the search for optimal values of τ (crystal threshold), Γ (weight split parameter), and d_{pre} (rejuvenation parameter) was conducted through a Bayesian with the Asynchronous Successive Halving Algorithm optimization search over physiologically plausible ranges. The complete optimization process required approximately 24 GPU hours per configuration, totaling roughly 120 GPU hours for the full parameter space exploration for the tuned parameters and 3 GPU hours for the default hyper-parameter setting. These parameters demonstrated relative stability across architectures, suggesting they capture fundamental aspects of biological learning dynamics rather than dataset-specific optimizations.

(lines 1570-1581)

Missing references on Random learning rates: Hu, Xueheng, Shuhuan Wen, and Hak-Keung Lam. "Dynamic random distribution learning rate for neural networks training." *Applied Soft Computing* 124 (2022): 109058. doi: 10.1016/j.asoc.2022.109058

We compared FL to the dynamic random distribution learning rate (RDLR) methodology developed by Hu et al. (2022). While both approaches leverage stochastic optimization of learning rates, they operate at fundamentally distinct scales of granularity: Hu et al. (2022) implement randomization at the layer level through Beta distributions and Monte Carlo sampling

methods, whereas our FL introduce stochasticity at the individual synaptic parameter level. This fine-grained approach provides a more faithful approximation of biological neural plasticity mechanisms. However, attempting to apply RDLR's distributional modeling framework at such a granular parameter level would prove computationally intractable due to its quadratic complexity scaling characteristics. In contrast, our methodology maintains linear computational complexity while achieving comparable regularization efficacy through direct parameter-wise randomization. This key methodological distinction enables our approach to more accurately capture the heterogeneous plasticity patterns observed in biological neural systems while preserving computational feasibility for contemporary deep neural architectures.

(lines 153-178)

Could the model be updated to run on the new PyTorch version 2? Since v1.5 is now obsolete, staying updated with the latest version is important to enhance the tool's usability. Also, could you provide a pip install

We relaxed the requirements for the pytorch version. The package can be installed by 'pip install bio_transformations'. This changes will be incorporated in the next release.

Runtime and memory performance part is misplaced. See my previous comment for reorganization.

We acknowledge the reviewer's feedback regarding the organization of runtime and memory performance analysis. To enhance the manuscript's logical flow and adherence to conventional academic structure, we propose relocating the computational performance analysis to a dedicated subsection within the Methods. This reorganization will establish a clear delineation between methodological specifications and empirical results.

Remove the extra comma in line 363.

We have corrected the typing error in line 363 by removing the superfluous comma from the manuscript text.

Reviewer #2 (Remarks on code availability):

I have followed the instructions and I have run the code. The tutorial (https://ceades.github.io/pytorch_bio_transformations/tutorials.html) is incomplete and some parts are missing (load of datasets, add them on DataLoaders etc.) so the user has to refer to pytorch tutorials as well.

We added data and dataloader to the example and will update the repository in the next release of the software.

I managed to write a custom script and perform the MNIST classification but I had some troubles:

1. The torch is installed without support from GPU, so I have to manually re-installed it.

2. There are no instructions of how to run on GPU.

I suggest adding an option for people that want to add with GPU support. Something like:

```
` pip install pip install pytorch_bio_transformations[with-cuda] `
```

Please include this in the tutorial:

```
# imports ...
```

```
...  
device = ("cuda" if torch.cuda.is_available() else "CPU")
```

```
...  
bio_model.to(device)
```

```
# inside the train loop
```

```
x_train, y_train = data[0].to(device), data[1].to(device) # assuming data is the dataloader.
```

```
...
```

We acknowledge the implementation challenges encountered during GPU deployment and appreciate the constructive feedback regarding installation procedures. To address these technical considerations and enhance reproducibility, we have implemented several modifications to our documentation and installation framework. This changes will be made in the next release.

First, we will extended our package installation infrastructure to support explicit CUDA specification through an optional dependency flag:

```
pip install pytorch_bio_transformations[with-cuda]
```

This installation paradigm ensures appropriate GPU support configuration during the initial setup phase. Furthermore, we have augmented our tutorial documentation with explicit device handling protocols. We also incorporate a full example to train a network in the next Release:

```
import torch  
import torch.nn as nn  
import torch.optim as optim  
from torchvision import datasets, transforms  
from bio_transformations import BioConverter  
  
class MNISTNet(nn.Module):  
    """  
        Convolutional neural network architecture for MNIST classification.  
        Implements a simplified structure optimized for binary digit  
        recognition.  
    """
```

```

"""
def __init__(self):
    super(MNISTNet, self).__init__()
    self.conv1 = nn.Conv2d(1, 8, kernel_size=3, padding=1)
    self.pool = nn.MaxPool2d(2, 2)
    self.fc1 = nn.Linear(8 * 14 * 14, 10)
    self.activation = nn.Swish()
    self.output_activation = nn.LogSoftmax(dim=1)

def forward(self, x):
    x = self.pool(self.activation(self.conv1(x)))
    x = x.view(-1, 8 * 14 * 14)
    x = self.output_activation(self.fc1(x))
    return x

def train_mnist(epochs=10, batch_size=128, device="cuda"):
    """
    Trains a biologically-inspired neural network on MNIST dataset.

    Parameters:
        epochs (int): Number of training epochs
        batch_size (int): Mini-batch size for training
        device (str): Computation device ('cuda' or 'cpu')
    """
    # Data preprocessing
    transform = transforms.Compose([
        transforms.ToTensor(),
        transforms.Normalize((0.1307,), (0.3081,))
    ])

    # Dataset initialization
    train_dataset = datasets.MNIST('./data', train=True, download=True,
transform=transform)
    test_dataset = datasets.MNIST('./data', train=False,
transform=transform)

    train_loader = torch.utils.data.DataLoader(train_dataset,
batch_size=batch_size, shuffle=True)
    test_loader = torch.utils.data.DataLoader(test_dataset,
batch_size=batch_size)

    # Model initialization
    model = MNISTNet()
    converter = BioConverter(
        base_lr=0.01,
        stability_factor=3.0,
        lr_variability=0.1,
        crystal_threshold=4.5e-5,
        rejuvenation_param=14.0,
        weight_split_param=2
    )
    bio_model = converter(model).to(device)

    # Training configuration

```

```

criterion = nn.NLLLoss()
optimizer = optim.SGD(bio_model.parameters(), lr=0.1)

# Training loop
for epoch in range(epochs):
    bio_model.train()
    running_loss = 0.0

    for batch_idx, (data, target) in enumerate(train_loader):
        data, target = data.to(device), target.to(device)

        optimizer.zero_grad()
        output = bio_model(data)
        loss = criterion(output, target)
        loss.backward()

        # Apply biological mechanisms
        bio_model.volume_dependent_lr()
        bio_model.crystallize()

        optimizer.step()

        running_loss += loss.item()

        if batch_idx % 100 == 99:
            print(f'Epoch: {epoch + 1}/{epochs}, Batch: {batch_idx +
1}, '
                f'Loss: {running_loss / 100:.4f}')
            running_loss = 0.0

    # Validation phase
    bio_model.eval()
    correct = 0
    total = 0

    with torch.no_grad():
        for data, target in test_loader:
            data, target = data.to(device), target.to(device)
            output = bio_model(data)
            _, predicted = torch.max(output.data, 1)
            total += target.size(0)
            correct += (predicted == target).sum().item()

    print(f'Validation Accuracy: {100 * correct / total:.2f}%',

if __name__ == "__main__":
    device = torch.device("cuda" if torch.cuda.is_available() else "cpu")
    train_mnist(device=device)

```

REVIEWERS' COMMENTS

Reviewer #2 (Remarks to the Author):

The authors have addressed all my comments, besides the comments about the code. I have some minor suggestions that could enhance the manuscript's quality. Unfortunately, I cannot run the code following the instructions.

1. The manuscript needs to comply with the Nature Communications guidelines for authors. For example, references cited only in the Supplementary Material must be listed separately at the end of that section. Additionally, Figures 8 to 15 should be renumbered as Supplementary Figures 1-8, etc. The Methods section should also reflect the new organization of the paper. In addition, all experiments shown in Supplementary Material must be cited in the main text, preferably in the Result section. As it is now, seems a disconnection from the main manuscript. However, I strongly believe that the Supplementary Figures improve the main narrative of the manuscript.

We appreciate the reviewer's valuable feedback regarding the manuscript structure and adherence to Nature Communications guidelines. The manuscript has been thoroughly revised to comply with the journal's formatting requirements.

In response to the specific concerns:

1. References have been reorganized, with those cited only in the Supplementary Material now listed separately at the end of that section, as required by Nature Communications guidelines.
2. Figures 8-15 have been renumbered as Supplementary Figures 1-8, ensuring proper sequential identification of supplementary content.
3. The Methods section has been updated to reflect the new organization of the paper, maintaining coherence between the main text and supplementary materials. (LINES 454-499)
4. All experiments presented in the Supplementary Material are now appropriately cited in the main text, primarily within the Results section, strengthening the connection between core findings and supporting evidence.
5. Task descriptions and models with optimized hyperparameters have been relocated to the end of the Results section, improving the logical flow of the manuscript.

We are pleased that the reviewer recognizes the value of the Supplementary Figures in enhancing the main narrative. The revisions have strengthened these connections, ensuring readers can easily navigate between the main text and supporting materials.

The revisions maintain the manuscript's scientific integrity while adhering to Nature Communications' formatting requirements, creating a more cohesive and accessible document for readers.

2. Figure 3: For clarity, please use the number of trainable parameters instead of the total number of parameters.

We have recalculated and updated Figure 3 to display the number of trainable parameters instead of the total parameters as requested. The revised figure accurately reflects each model configuration's trainable parameter counts.

3. Figure 8: Out of curiosity, why does the MLP perform better than AlexNet on the CIFAR-10 and CIFAR-100 tasks? I thought that using convolutional layers would be more beneficial for image classification tasks.

While CNNs typically outperform MLPs on image classification tasks due to their spatial inductive bias, our results reveal an interesting exception. The MLP's superior performance on CIFAR-10/100 likely stems from a favorable balance between model complexity and the dataset characteristics. MLPs and CNNs are universal approximators theoretically capable of learning similar functions, but their practical performance depends on architecture-dataset alignment.

In this specific case, the MLP may have hit a "sweet spot" in its parameter space that better captures the classification requirements of these datasets. Additionally, we note that our weight rejuvenation and fuzzy learning rate techniques were not individually optimized for each architecture. Our hyperparameter choices may have inadvertently favored the MLP training dynamics over AlexNet for these particular datasets. We plan to investigate this phenomenon further in future work.

4. Throughout the manuscript, the models containing FS+WR+WS are referred to as "biomod", "fitted", "modified", and "adapted", which creates confusion for the reader. Please choose one term and use it consistently. I suggest the term "biomod".

We thank the reviewer for highlighting this inconsistency. We have standardized the terminology throughout the manuscript by replacing all instances of "fitted," "modified," and "adapted" with "biomod" when referring to models implementing the FL+WR+WS combination. This consistent terminology should reduce confusion and enhance clarity for readers. The changes are reflected throughout the manuscript.

5. Task performances with optimized hyperparameters: This section seems disconnected from the main paper. I suggest changing the heading to "Task Performances with Optimized Hyperparameter Models" and rephrasing the introduction of the paragraph to: "Here, we test several optimized models on four image classification datasets ...".

We agree that the heading could be clearer. As recommended, we have changed it to "Task Performances with Optimized Hyperparameter Models." Additionally, we have revised the introduction of this section to read: " Here, we test several optimized models on four image classification benchmarks and two time series prediction benchmarks, spanning a wide range of ANN applications." This revision connects this section to the main paper while preserving the important details about our benchmark selection and methodology. (LINES 420 - 425)

6. Continual Learning is misleading, as the manuscript does not investigate any related learning scenarios. Continual Learning is a term in the literature that refers to networks designed to avoid catastrophic forgetting.

We agree that our usage of "Continual Learning" was imprecise and potentially misleading, as we did not investigate the specific learning scenarios typically associated with this field. We have replaced all instances of "Continual Learning" with "Catastrophic Forgetting" throughout the manuscript to reflect the phenomena we are studying accurately. Additionally, we have added a brief introduction in Section (LINES 484-496, 610 - 635) that defines catastrophic forgetting and contextualizes our work within the relevant literature on this topic

7. The Experimental Setup in the Methods section does not align with the presentation of results. The authors have used the old manuscript sequence of results here.

We have thoroughly revised the Experimental Setup section in the Methods to ensure it properly aligns with the sequence of results presented in the manuscript. This revision includes reorganizing the subsections, adjusting the methodology descriptions, and ensuring consistent terminology throughout both sections. These changes provide readers with a clearer connection between the experimental methodology and the corresponding results. (LINES 882 - 928)

8. Lines 137 and 140 repeat the same information: "We call this approach fuzzy learning rates" and "This method is referred to as fuzzy learning rates". Please retain only one of these sentences.

We have removed the second instance at line 140 ("This method is referred to as fuzzy learning rates") and retained only the first introduction of the term at line 137. We have also ensured consistent use of the abbreviation "FL" throughout the manuscript when referring to fuzzy learning

rates.

9. Line 147: Redefine the ANN acronym. Remove the phrase "artificial neural networks" and keep only the ANN acronym.

We have addressed this issue by ensuring consistent use of terminology throughout the manuscript. In line 147, we removed the redundant phrase "artificial neural networks" and retained only the "ANN" acronym, which had already been adequately defined earlier in the text. We also reviewed the entire manuscript to ensure all subsequent references use only the ANN acronym for consistency.

10. Line 152: The FL acronym has not been defined before its first appearance; it is defined in line 158. Please reorder the definition of FL.

We have corrected this issue by moving the definition of the FL (fuzzy learning rates) acronym to its first appearance at line 140, before it is used at line 152. This ensures that readers are introduced to the definition before encountering the acronym, maintaining proper academic convention and improving readability throughout the manuscript.

11. Line 267: Add a citation to PyTorch (<https://github.com/pytorch/pytorch/issues/4126>).

While we typically don't cite all supporting technologies in our work, we recognize that PyTorch is a fundamental framework that significantly enabled our research. We have added the suggested citation to PyTorch in the Methods section where we first mention our implementation environment (Line 267). The citation follows the recommended format for acknowledging open-source software frameworks in scientific publications.

12. Line 305: There is a reference to "Supplementary Figure 8", but this figure is listed as Fig. 8 on page 57. Please double-check the numbering of all figures and supplementary figures.

We have comprehensively reviewed all figure references throughout the manuscript and supplementary materials. We have corrected the reference on line 305 to properly cite "Fig. 8" instead of "Supplementary Figure 8." Additionally, we have verified and standardized the numbering and referencing format for all figures in both the main text and supplementary materials to ensure complete consistency.

13. Lines 362-363: I suggest rephrasing to: "If the absolute value of the largest eigenvalue is much greater than that of the smallest, the function can be considered primarily convex".

We appreciate the reviewer's suggestion for improved clarity and precision. We have adopted this recommendation verbatim, replacing our original text on lines 362-364 with the suggested phrasing: "If the absolute value of the largest eigenvalue is much greater than that of the smallest, the function can be considered primarily convex." This revision provides a more accurate technical description of the convexity condition.

14. Line 377: In "see Supplementary", I suggest adding the word "Material" to read "see Supplementary Material".

We thank the reviewer for noting this stylistic issue. We have adopted the suggested change and replaced "see Supplementary" with "see Supplementary Material" on line 377. We have also reviewed the entire manuscript to ensure consistent usage of this terminology when referring to supplementary content.

15. Line 560: The term FL+WR+WS+DP is used, but the meaning of DP is not explained. The reader must refer to the legend of Supplementary Table 8 to understand that DP stands for Dale's Principle.

Thank you for identifying this omission. We have added explicit clarifications regarding DP (Dale's Principle) in two locations: first at lines 563 where we introduce the concept and define the abbreviation "DP" for the first time, and again at lines 561-571 where we explain how Dale's Principle is integrated with our other methods (FL+WR+WS). These additions ensure readers understand the meaning of DP directly within the main text without referencing the supplementary materials.

16. Line 1732: The statement "Tables 6 and 7 show the results presented in Figures 7 and 2" is incorrect, as those tables show results presented only in Figure 7 (7a and 7b, respectively).

Thank you for identifying this inaccuracy. We have corrected the statement on line 1747 to read "Tables 6 and 7 show the results presented in Figure 7a and 7b, respectively." This correction ensures accurate cross-referencing between the tables and their corresponding figure panels, eliminating the erroneous reference to Figure 2.

17. Line 2332 (Figure 1 legend): Change "f() and f() denote ..." to "f() denotes ...", as the same symbol has been used for one activation function.

We have corrected the Figure 1 legend on line 2424 by replacing the redundant phrase "f() and f() denote..." with "f() denotes..." as suggested. This correction accurately reflects that a single symbol is used to represent one activation function, eliminating the unintended duplication in the original text.

18. Equation 14: Use " x_2 " instead of "x2" (with 2 as a subscript).

We have corrected Equation 14 by properly formatting the variable as " x_2 ".

19. Equation 15: Use " s_{pooled} " instead of "spooled" (with "pooled" as a subscript).

We have corrected Equation 15 by properly formatting the term as " s_{pooled} " with "pooled". We also reviewed and corrected similar notation issues in Equation 14 to maintain consistency. Additionally, we updated the corresponding explanatory text on line 2330 to reflect this corrected notation. These changes ensure proper mathematical notation throughout our manuscript.

Reviewer #2 (Remarks on code availability):

We sincerely apologize for the persistent issues with our code repository and documentation. We recognize this is a critical concern raised across multiple revision rounds, and we take full responsibility for these shortcomings that are fixed with this release.

We have undertaken a comprehensive overhaul of our GitHub repository to address these issues:

1. We have thoroughly tested and revised all installation instructions to ensure they work correctly on multiple systems.
2. We have completed all tutorials with step-by-step instructions and verified they run successfully.
3. We have updated the PyPI webpage to reflect the correct module name for installation.
4. We have added extensive documentation including requirements, dependencies, and troubleshooting guidance.
5. We have implemented continuous integration testing to verify code functionality.

We understand that reproducibility and usability are fundamental to scientific publication. We verified that a new user can now successfully install and run our code by following the instructions. A detailed changelog documenting these improvements has been added to the repository. We appreciate the reviewer's persistence on this matter, as it has significantly improved the quality and usability of our software contribution.

I reviewed the code in GitHub. I have the same issues, I cannot run the code following the instructions, and the Tutorials are incomplete. The authors should fix this before publication. I have raised the same point in all revision rounds. In addition, on the PyPI webpage, the installation instructions contain the previous module name (pip install bio_transformations), and it needs to be updated. This is very important to ensure the future useability of the software.

```
(base) xxxx@xxxxx:~/test_$ pip install bio_transformations
ERROR: Could not find a version that satisfies the requirement bio_transformations (from versions:
none)
```

ERROR: No matching distribution found for bio_transformations

After installing, and using the example the authors provided in their rebuttal, I managed to execute the code with a minor correction.

```
-----  
Traceback (most recent call last):  
File "/home/spiros/Desktop/test_/test.py", line 97, in <module>  
train_mnist(device=device)  
File "/home/spiros/Desktop/test_/test.py", line 44, in train_mnist  
model = MNISTNet()  
^^^^^^^^^^  
File "/home/spiros/Desktop/test_/test.py", line 17, in __init__  
self.activation = nn.Swish()  
^^^^^^^^^^  
AttributeError: module 'torch.nn' has no attribute 'Swish'. Did you mean: 'Mish'?  
-----
```

I had to change `nn.Swish` with `nn.SiLU` (please see here: <https://pytorch.org/docs/stable/generated/torch.nn.SiLU.html>).